# STAR: A Benchmark for Astronomical Star Fields Super-Resolution

Guocheng Wu[1] [*][†]   Guohang Zhuang[1,2] [*][†]

Jinyang Huang[2]   Xiang Zhang[3]   Wanli Ouyang[4]   Yan Lu[1,4] [‡]

[1] Shanghai Artificial Intelligence Laboratory   [2]Hefei University of Technology

[3]University of Science and Technology of China   [4]The Chinese University of Hong Kong

## Abstract

Super-resolution (SR) advances astronomical imaging by enabling cost-effective high-resolution capture, crucial for detecting faraway celestial objects and precise structural analysis. However, existing datasets for astronomical SR (ASR) exhibit three critical limitations: flux inconsistency, object-crop setting, and insufficient data diversity, significantly impeding ASR development. We propose **STAR**, a large-scale astronomical SR dataset containing 54,738 flux-consistent star field image pairs covering wide celestial regions. These pairs combine Hubble Space Telescope high-resolution observations with physically faithful low-resolution counterparts generated through a flux-preserving data generation pipeline, enabling systematic development of field-level ASR models. To further empower the ASR community, **STAR** provides a novel Flux Error (FE) to evaluate SR models in physical view. Leveraging this benchmark, we propose a Flux-Invariant Super Resolution (FISR) model that could accurately infer the flux-consistent high-resolution images from input photometry, suppressing several SR state-of-the-art methods by 24.84% on a novel designed flux consistency metric, showing the priority of our method for astrophysics. Extensive experiments demonstrate the effectiveness of our proposed method and the value of our dataset. Code and models are available at `https://github.com/GuoCheng12/STAR`

## 1   Introduction

Image quality is critical to astronomical observation, while high quality means finer astrophysical structures and enables precise measurements [1, 2, 3]. This results in the astronomy community always establishing new telescopes to seek high-quality and high-resolution surveys, even facing high costs [4, 5]. Different from astronomy, in natural image processing, the software computer vision Super Resolution (SR) technique [6, 7, 8]has provided a series of successful methods to achieve high-quality and high-resolution observations in an economical way [9, 10]. So, there is obviously an opportunity to introduce the computer vision SR method to process high-quality astronomical images. However, there remains a great challenge – data.

Existing datasets [11, 12] in astronomical super resolution (ASR) have 3 drawbacks: physically trivial, object-centric, and limited-scale. 1). **Flux Inconsistency**: In the real world, telescopes under different observation resolutions have a flux consistency relation [13, 14]. Specifically, although a celestial object has different levels of distortions under low resolutions, it almost has the same total flux as in high resolutions because of the telescope imaging principle [13, 15]. However, existing datasets

---

[*]Equal contribution.

[†]Work done during internship at Shanghai Artificial Intelligence Laboratory.

[‡]Corresponding authors. Email: luyan@pjlab.org.cn

39th Conference on Neural Information Processing Systems (NeurIPS 2025) Track on Datasets and Benchmarks.

have significant drifts from this property because they directly use simple interpolation [16, 17], suitable for natural images but conflicts with astronomical observations. This catastrophic limitation makes existing datasets almost physically trivial, significantly affecting their scientific value. 2). **Object-Crop Configuration**: Each image in existing ASR datasets only contains a center-cropped and resized singular celestial object (e.g., stars or galaxies) [16, 18]. This ideal configuration neglects many valuable patterns beyond single object, important in astrophysics, such as large-scale structure [19], cross-object interaction [20], and weak lensing [21, 22], limiting the value of existing datasets. 3). **Insufficient Data Diversity**: The scale of existing ASR datasets ranges from 1,597 to 17,000 [11, 12, 16, 17, 18, 23]. The restricted scale limits the ability of the learned model and makes evaluation unreliable and unfair. To address the above-mentioned dataset limitations, we introduce a new dataset called **STAR**.

STAR is a **large-Scale** ASR dataset. It consists of 54,738 high-resolution star field images captured by the Hubble Space Telescope (HST) [24]. Each image is totally **field-level**, covering a large range of star fields and average containing 30 objects and complex scenarios including multiple celestial objects, cross-object interaction and weak lensing phenomenon, as Figure 1 shows. Compared with existing ASR datasets, STAR provides approximately at least 15 times more observation objects per image on average, while also offering 60% of cosmic information outside the object area (e.g, like diffuse interstellar medium (ISM) regions [25]), significantly showing the scale priority. We provide overall advantages of the STAR for other datasets in Tab. 1.

Except that, to tackle the 'Physical trivial' problem, STAR proposes a **flux-consistent** data generation pipeline, which processes cross-resolution image pairs fitting the aforementioned real telescope flux-consistent property, making the entire dataset physically faithful. Furthermore, STAR provides a novel Flux Error (FE) to evaluate SR models from a physical perspective, ensuring their outputs align with astrophysical principles critical for reliable scientific analysis.

With the STAR, we evaluate several state-of-the-art SR methods, including both natural [8, 26, 27, 28, 29, 30] and astronomical SR methods [11, 18] to quantify their generalization ability to the field-level ASR topic, noting that many astronomical SR methods directly adopt natural SR methods. Unfortunately, they cannot provide satisfactory results. We analyze that the main reason is the lack of specific optimization for the flux-consistency prior. Due to this, we propose a novel field-level ASR model, Flux-Invariant Super Resolution (FISR). It introduces the flux consistency property at both the model design and optimization views to fulfill the flux relationships neglected by previous ASR works. At the model view, FISR has a series of specific designs to extract flux information from low-resolution input as visual prompts following astrophysical ideas. These prompts are then injected into the model and give the ability to perceive input flux accurately, allowing the model to propagate consistent flux cues from low-resolution inputs to predicted high-resolution outputs. And at the optimization view, we provide a Flux consistency loss (FCL) which constrains the photometry gap for each celestial object between the ground-truths and predictions, highlighting the importance of flux during the model optimization process and leading to a more reliable trained model.

- **STAR Benchmark**: We introduce STAR, a large-scale, flux-consistent ASR benchmark with 54,738 cross-resolution image pairs from HST F814W star fields. Unlike prior datasets, STAR captures field-level complexity, offering 15 times more objects per image and 60% additional cosmic information, using a flux-consistent pipeline.
- **Flux Error (FE)**: We present FE, a novel metric to evaluate SR models' alignment with astrophysical flux conservation, ensuring reliable photometric analysis.
- **Flux-Invariant Super Resolution (FISR) Model**: We propose FISR, a field-level ASR model and a Flux Consistency Loss, outperforming existing methods by addressing flux relationships neglected in prior work.

## 2 Related Works

### 2.1 Super-Resolution Techniques

Super-resolution (SR) techniques, aimed at reconstructing high-resolution (HR) images from low-resolution (LR) inputs, have significantly advanced, offering critical tools for enhancing astronomical images. Traditional SR methods, including interpolation, deconvolution, and learning-based approaches like sparse representation [31], modeled image degradation or statistical relationships to

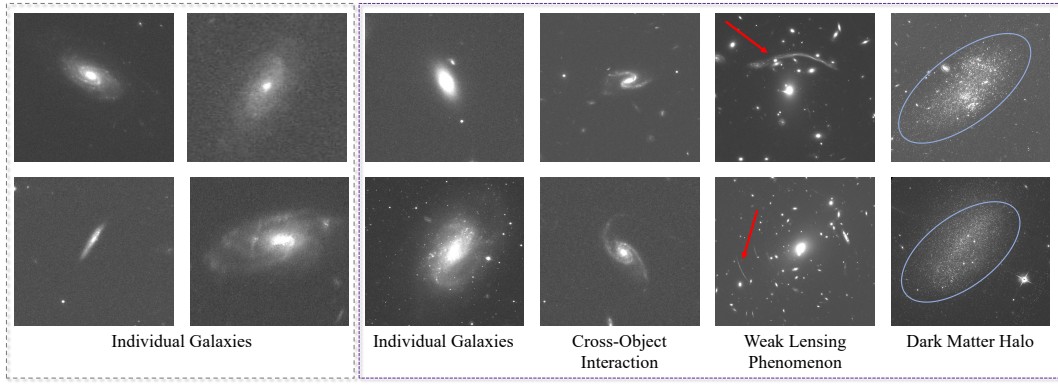

|  | Individual Galaxies | Individual Galaxies | Cross-Object Interaction | Weak Lensing Phenomenon | Dark Matter Halo |
| **(a) Previous Dataset Examples** | | **(b) STAR Dataset Examples** | | | |

Figure 1: Comparison of previous datasets and ours, highlighting richer structures such as cross-object interaction, weak lensing, and dark matter halos.

Table 1: Comparison of existing astronomical SR datasets.

| Dataset | Size | Type | Downsampling | Multiple Celestial | Flux Consistency |
| --- | --- | --- | --- | --- | --- |
| AstroSR [11] | 2000 | Galaxy | × | × | × |
| QQ Shan et al [12] | 9383 | Galaxy | × | × | × |
| W Song et al [16] | 1597 | Solar | ✓ | × | × |
| DiffLense [17] | 2880 | Galaxy | × | × | × |
| ZJ Luo et al [18] | 17000 | Galaxy | × | × | × |
| WJ Li et al [23] | 14604 | Galaxy | ✓ | × | × |
| STAR | 54738 | Star field | ✓ | ✓ | ✓ |

recover details, laying the foundation for SR applications. The advent of deep learning revolutionized SR, with convolutional neural networks (CNNs) enabling robust feature learning (e.g., [6, 8, 32]) and generative adversarial networks (GANs) enhancing perceptual quality through adversarial training (e.g., [33, 34, 35, 36]). Recent advancements introduced transformer-based models, leveraging global attention for superior detail recovery (e.g., [26, 29, 37, 29, 38]), and diffusion-based models, using iterative denoising for high-quality image generation (e.g., [36, 39, 40]). Unlike natural image SR, which prioritizes visual perception, astronomical SR must balance perceptual quality with the physical integrity of scientific data, as required in applications like stellar population analysis [10, 41].

## 2.2 Astronomical Image Super-Resolution

Super-resolution (SR) techniques tailored for astronomical images have evolved to address the unique challenges of celestial data, achieving notable success in enhancing specific targets like stars and galaxies. To improve image quality, the most direct method is to enhance the hardware capabilities of astronomical telescopes, leveraging advancements in optical and detection technologies. Common hardware improvements include increasing the telescope aperture, equipping telescopes with adaptive optics systems, advancing photodetector technology, and optimizing optical component design [42, 43, 44, 45, 46, 47]. These advancements complement software-based SR methods, which have significantly refined image resolution. Early software approaches, such as deconvolution [10, 41] and multi-frame stacking [48, 49], successfully improved the resolution of isolated stellar and galactic images by modeling point spread functions (PSFs) [47] or combining multiple exposures. These approaches enabled precise analyses of individual stars and galaxy morphologies [50]. More recently, computational advancements have explored SR for broader astronomical applications, primarily focusing on single-target scenarios like galaxies [51], Sun [52], X-ray sources (nebulae, active galactic nuclei, etc.) [53]. Our work extends this progress by developing a large-scale star field dataset that captures diverse astronomical conditions, enabling robust SR model training for complex star field scenes, an area previously underexplored. Additionally, we integrate flux consistency constraints to ensure reconstructed images preserve critical physical properties, enhancing their reliability for quantitative analyses such as photometry and stellar dynamics.

## 2.3 Flux Consistency in Astronomical Image Processing

Flux consistency, ensuring that the total light intensity (flux, or photons received per unit area) of celestial objects in processed images matches original observations, is a cornerstone of astronomical image analysis, underpinning reliable photometry and stellar population studies [54, 55]. Historical efforts prioritized flux consistency to preserve measurement accuracy in star clusters and galaxies. The modern space telescope SDSS also follows this principle [56]. However, the complexity of star fields, with diverse brightness and overlapping objects, poses ongoing challenges. Our work advances this field with STAR, a large-scale star field dataset ensuring flux-consistent image pairs, and novel flux consistency constraints, enhancing the scientific reliability of star field analyses.

## 3 STAR

Following natural SR works, we construct cross-resolution image pairs by downsampling high-resolution images. We choose Hubble Space Telescope (HST)[4] survey data as our high-resolution images due to its widely recognized data quality and rich historical data accumulation. Given a high-resolution HST image, we first apply a point spread function (PSF) [47] kernel to simulate the optical blurring effects caused by low-quality telescopes and atmospheric turbulence.

Next, we downsample the image by a factor of $s$ using a flux-conserving scheme. Finally, since both HR and LR images have large spatial dimensions, we divide them into smaller sub-images to facilitate model training. This pipeline generates physically consistent cross-resolution pairs of images, crucial for robust super-resolution model training.

### 3.1 High-resolution Data Collection

The HST is a space-based observatory designed to capture high-resolution astronomical images across a wide range of wavelengths, from ultraviolet to near-infrared. It provides two kinds of data, including `calibration` and `science`. Calibration data is used to correct instrumental effects while the science has a verified quality for scientific research. So we choose scientific data due to its high and reliable quality.

The `science` data consists of images captured by various imaging instruments onboard HST, such as the Advanced Camera for Surveys (ACS) [57], Wide Field Camera 3 (WFC3) [58], and Wide Field and Planetary Camera 2 (WFPC2) [59]. We selected the ACS Wide Field Channel (WFC/ACS) for its high sensitivity in optical wavelengths (350–1050 nm) and the widest field of view (202" × 202"),

which are critical for capturing high-resolution images of extended astronomical objects.

Astronomical data are captured under different filters, like natural images under Red, Green and Blue filters. Here, for the WFC/ACS `science` data, we keep the F814W filter (centered at 814 nm, also known as the I-band) data due to the band of the F814W is widely used in star field studies because its wavelength (centered at 814 nm) effectively resolves individual stars in crowded fields thile maintaining high photmetric accuracy [60]. So we choose it for its representative.

HST observes one location many times, resulting in a large number of overlapping images. To remove high-overlapping data but keep diversity as wide as possible, we use the farthest point sampling strategy [61] on the HST covered celestial regions and finally select 70 representative wide field images covering an extremely large range of celestial regions but without any overlapping.

### 3.2 PSF Blurring

Different resolution devices share different PSF blurring. To simulate this phenomenon, we adopt two representative PSF models. The first is the Gaussian PSF [62], which is widely used to approximate blur caused by atmospheric turbulence or instrumental imperfections. The second is the Airy PSF, which is a device-specific kernel widely used in astrophysics, describing the instrumental effect that occurs when a telescope resolution changes. Their detailed formulation could be seen in the supplementary. With these two PSFs, we define two blurring settings: one is a single Gaussian kernel $G$, and the other is a combination of Airy and Gaussian to simulate more complex scenarios.

---

[4] https://www.stsci.edu/hst

After operating PSF blurring, we perform a flux consistency downsampling scheme to obtain realistic low-resolution counterparts, which we detail in the next section.

### 3.3 Flux Consistency Downsampling

The flux consistency relation in real-world telescopic observations stems from the telescope imaging principle. The value of each pixel in observational data corresponds to the photon flux captured by its corresponding CCD pixel. Consequently, when imaging the same celestial region, a single CCD pixel in lower-resolution instruments covers a larger spatial area—equivalent to integrating photons from multiple high-resolution CCD pixels. This mechanism enables Flux-Consistent Downsampling by calculating the celestial receptive field ratio of each pixel across resolution scales. The details of this flux consistency downsampling could be seen in the supplementary.

### 3.4 HR Image Subdivision

The previous flux consistency downsampling scheme provides us a flux-consistent image pairs. To further optimize model training effectiveness, we divide the HR and LR wide field images into smaller sub-images. Because astronomical images contain some outlier regions (have NaN values) due to geometric calibration in DrizzlePac[63] processing, we retain only patches with >80% valid regions containing stellar features. As a result of these processing steps, we obtain a large set of high-quality HR-LR image pairs and construct the STAR dataset for training and evaluating astronomical super-resolution models.

### 3.5 Flux Error

The STAR provides a Flux Error (FE) to evaluate flux consistency between a ground truth and its corresponding prediction. The FE measures the flux value gap for each object, so its computation process is based on astrophysical photometry. The basic process of the photometry is deriving flux by detection. We follow this idea. For a given ground-truth and predicted image pair, we compute the FE as following two steps: **1)**. For the ground-truth image, we use the Starfinder toolkit [64] to detect celestial objects and obtain their parameters. Then, we derive the flux of each object by a widely-used elliptical photometry method [64]. **2)**. For predicted images, we do not operate object detection but directly use detection results from the ground-truth image because they provide reliable object catalogs covering both strong and weak sources. The following photometry is the same as the ground truth. After these two steps, we have a two set of flux values, denoted as $\{v_{pred}^1, v_{pred}^2 ..., v_{pred}^N\}$ and $\{v_{gt}^1, v_{gt}^2 ..., v_{gt}^N\}$ where $N$ is the number of detected objects. The FE is computed by the following:

$$\text{FE} = \frac{1}{N} \sum_{i=1}^{N} \left| v_{gt}^i - v_{pred}^i \right|. \tag{1}$$

Lower FE means higher flux consistency. Since the flux value is related to the object shape and the pixel flux in object regions, this metric could reflect the geometric shape consistency for each object in the reconstruction image, physically informed flux consistency and weak source object reconstruction quality simultaneously.

## 4 Method

### 4.1 Overview

The entire pipeline of our Flux-Invariant Super-Resolution (FISR) is shown in Fig. 2. The low-resolution input image are sent into two paths. The first one is an encoder block consisting of a convolution operation and multiple transformer blocks, which represent the input image as multi-scale feature maps. The second is a novel Flux Guidance Generation (FGG) module that extracts flux information as multi-scale flux guidance representations. Each flux guidance is sent into a scale-wise Flux Guidance Controller (FGC) to enhance the encoded feature maps. This scheme highlights regions with significant flux to guide the network's focus toward astrophysically relevant structures. Then, multiple decoder transformer blocks progressively process these enhanced features and finally upsample them as the output images.

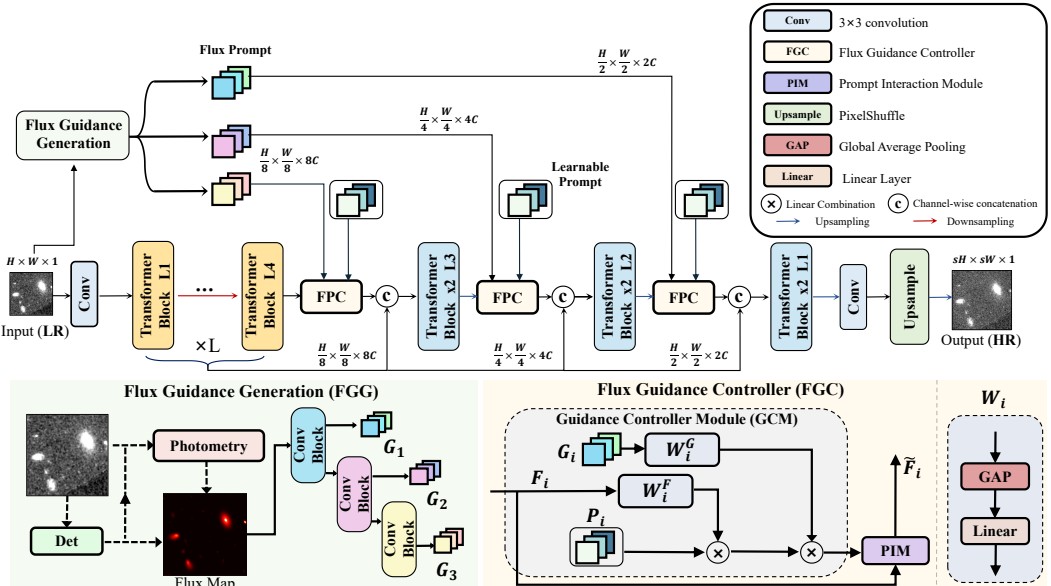

Figure 2: Overview of the FISR approach. The input image is processed through an encoder branch and a Flux Guidance Generation (FGG). Flux information is extracted via FGG and injected into the Flux Guidance Controllers (FGC) to enhance the encoded feature map. The enhanced features are then decoded and upsampled to produce the final high-resolution output..

## 4.2 Flux Guidance Generation

We propose the (Flux Guidance Generation) FGG module to introduce the flux information. Specifically, given an input image, FGG first represents flux information for every celestial object in the input image into a flux map, then transfers the flux map as multi-scale features as guidance which will then be used in the next FGC module to enhance multi-scale feature maps.

To generate the flux map, FGG also computes flux by detection. It detects celestial objects first and obtains a bounding box for each object. Then the photometry process is operated to derive object fluxes. The entire detection and photometry process is real-time. With these, FGG generates the map following the idea of drawing bounding boxes on a white background and corresponding flux value in each bounding box region.

In practice, because the 'bounding box' of a celestial object is essentially a rotatable ellipse rather than a rectangle in standard object detection, the drawing scheme FGG has some modifications. Specifically, for each ellipse, FGG puts a rotatable Gaussian Kernel at the center location and adjusts the Gaussian standard deviation based on the ellipse size. Then, FGG multiplies the object flux value directly with the rotated Gaussian kernel to modulate the kernel value. Finally, FGG draws all these kernels together as the final flux map.

Finally, a multi-scale convolutional block transforms the flux map into a corresponding flux guidance representation. In our architecture, we employ three such blocks at different feature levels to generate a hierarchy of flux guidance, denoted as $\{\mathbf{G}_1, \mathbf{G}_2, \mathbf{G}_3\}$. The level 3 is set to align the block number in the decoder, following common setting in a popular SR baseline [27].

## 4.3 Flux Guidance Controller

Based on the flux guidance produced by the FGG module, the (Flux Guidance Controller) FGC interacts such guidance with the encoder features, generating the enhanced features. The FGC is scale-wised and for the $i$-th scale FGC, its interaction pipeline is represented as follows:

$$\hat{\mathbf{F}}_i = \text{PIM}_i \left( \text{GCM}_i \left( \mathbf{P}_i, \mathbf{F}_i, \mathbf{G}_i \right), \mathbf{F}_i \right). \tag{2}$$

It shows that the enhanced feature $\hat{\mathbf{F}}_i \in \mathcal{R}^{H \times W \times C}$ is derived from a guidance-controlled feature and an original feature $\mathbf{F}_i \in \mathcal{R}^{H \times W \times C}$ by a prompt interaction module function $\text{PIM}(\cdot, \cdot)$, which is set

same as combines image features with guidance-controlled feature and dynamically adjusts the input features through a transformer block. following PromptIR [27]. The guidance-controlled features are computed by a guidance controller module function $\text{GCM}(\cdot, \cdot, \cdot)$ that takes a learnable prompt $\mathbf{P}_i$, the flux guidance component $\mathbf{G}_i \in \mathcal{R}^{H \times W \times C}$ and the encoded feature $\mathbf{F}_i$ as inputs. The learnable prompt $\mathbf{P}_i \in \mathcal{R}^{H \times W \times C \times K}$ contains $K$ learnable patterns expected to represent blind property in the image restoration process [27]. The detail of GCM is as follows:

$$\text{GCM}_i\left(\mathbf{P}_i, \mathbf{F}_i, \mathbf{G}_i\right) = \sum_{k \in K} W_i^F(\mathbf{F}_i) \odot W_i^G(\mathbf{G}_i) \odot \mathbf{P}_i, \tag{3}$$

where $W_i$ means learnable modules, consisting of a global average pooling and a linear layer, takes input corresponding features and derives a weight with the size of $1 \times 1 \times 1 \times K$ to indicate the mportance of the learnable prompt patterns. $\odot$ means Hadamard product with dimension broadcast while $\sum_{k \in K}$ means sum the last dimension of the multiplied features with the size of $H \times W \times C \times K$ to derive the final output features with the size of $H \times W \times C$.

### 4.4 Flux Consistency Loss

The idea of the Flux Consistency Loss is to train the model to generate flux consistent prediction. We propose a simple yet effective scheme to achieve this goal by highlighting flux-related regions. We first operate the flux map generation scheme described in Section 4.2 on the ground-truth image, denoted as $M$. Then we use this map to weighted the pixel wised supervision as follows:

$$\mathcal{L}_{\text{flux}}(I_{\text{pred}}, I_{\text{gt}}) = \sum_{x,y} M(x, y) \cdot |I_{\text{pred}}(x, y) - I_{\text{gt}}(x, y)|. \tag{4}$$

Combined it with a reconstruction loss (L1 or L2), the total loss is:

$$\mathcal{L}_{\text{total}} = \mathcal{L}_{\text{recon}}(I_{\text{pred}}, I_{\text{gt}}) + \lambda \cdot \mathcal{L}_{\text{flux}}(I_{\text{pred}}, I_{\text{gt}}), \tag{5}$$

where $\lambda$ balances terms. This loss takes into account both traditional regression supervision and flux consistency constraints, which brings more physically reliable ASR results.

## 5 Experiment

### 5.1 Experimental Setup

**Dataset**. We use the proposed STAR to process comparisons, evaluate our model and perform ablation studies. The downsampling ratio $s$ is set as 2 and 4, respectively. As mentioned before, the blurring PSF has two settings: Gaussian only and Gaussian+Airy. The experimental results of the latter are placed in Appendix C.

**Implementation details**. Each model is trained for 100 training epochs with a batch size of 16 on 8 H800 GPUs. The initial learning rate is set to 2e-4 and decayed by a factor of 0.01 at the 50th epoch. To make the training more stable, we apply the linear warm-up strategy in the first epoch. More details about the environment setting can be found in Appendix F.

**Evaluation protocols**. Following prior work, we adopt Peak Signal-to-Noise Ratio (PSNR) [65] and Structural Similarity Index (SSIM) [66] as standard evaluation metrics to measure reconstruction quality. Further, we use the proposed FE to evaluate physical fidelity to quantify the alignment between the output of the model and astrophysical principles. Note that we do not use SR commonly-used perceptual metrics, such as LPIPS [67], which leverages ImageNet [68] pre-trained deep models [28, 69] to measure semantic similarity. Because in our ASR task, they are not suitable due to the significant domain gap between the pre-trained domain and astronomical images.

### 5.2 Quantitative and Qualitative Results

We compare our method with several methods in Tab. 2, including state-of-the-art natural SR methods, such as HAT [29] and classical SR methods, for example, SwinIR [26]. SwinIR and RealESRGAN [34] are also popular in the ASR topic [11, 17], which is also an important factor that we choose them to compare.

Table 2: Performance comparison of different methods under ×2 and ×4 super-resolution. Evaluation metrics include PSNR, SSIM, and Flux Error.

| Scale | Metric | Bicubic | EDSR [8] | RealESRGAN [34] | RCAN [32] | SwinIR [26] | HAT [29] | FISR (ours) |
|---|---|---|---|---|---|---|---|---|
| ×2 | PSNR↑ | 28.6021 | 35.3816 | 36.8363 | 36.3703 | 37.2205 | 37.2501 | **37.8779** |
| | SSIM↑ | 0.6842 | 0.8054 | 0.8225 | 0.8240 | 0.8286 | 0.8295 | **0.8311** |
| | FE↓ | 4.9418 | 1.4623 | 7.3632 | 0.9237 | 0.813 | 0.7636 | **0.5739** |
| ×4 | PSNR↑ | 26.0518 | 33.8736 | 34.3725 | 34.9823 | 34.6655 | 34.9142 | **35.1788** |
| | SSIM↑ | 0.6005 | 0.7201 | 0.7223 | 0.7263 | 0.7258 | **0.7276** | 0.7266 |
| | FE↓ | 7.8733 | 1.3841 | 4.0782 | 1.0550 | 1.0657 | 1.0256 | **1.0125** |

From Tab. 2, it could be shown that our approach outperforms existing methods across different evaluation criteria. Compared with HAT [29], our method achieves complete priority under the ×2 case while keeping most advantages under the harder ×4 case, showing the effectiveness of our approach under different scenarios. The better PSNR and SSIM demonstrate higher reconstruction quality of our FISR model. What's more, a significant FE priority shows the satisfactory physical reliability of our method, proving its value in physically faithful high-precision astrophysics image processing. The comparison of the second-best SwinIR [26] who is the current state-of-the-art ASR method, demonstrates that our method achieves a new state of the art in the ASR topic, further proving our effectiveness.

### 5.3 Ablation Study

To evaluate the contribution of each proposed component in our FISR framework, we conduct comprehensive ablation studies on the STAR dataset under the ×2 super-resolution setting. Tab. 3 presents the performance changes when incorporating or removing the Flux Guidance Generation (FGG), Flux Guidance Controller (FGC) and Flux Consistency Loss (FCL).

The ablation of FCL is straightforward, but it is not easy and also trivial for us to ablate FGC and FGG solely. So here, we modify the ablation study of FGC and FGG as the ablation of FG-Modules and Flux cues. FG-Modules means keeping all learnable modules of FGG+FGC, but replacing the flux map of FGG as the original input image. Only using FG-Modules means ablating the input flux cues. The idea behind this design is that the key motivation of FGC+FGG is to introduce input flux information. So, the ablation of the flux map is effective because it could demonstrate that the gains introduced by the FGC and FGG are actually from flux information rather than more parameters.

**Effect of FCL:** In the $1^{st}$ line of Tab. 3, we show the performance of our baseline, PromptIR [27]. In the $2^{nd}$ line, we introduce the FCL, significantly decreasing the physical faithful metric FE about 0.06+, showing its function to achieve flux consistency. Similar gains can be found by comparing the $5^{th}$ and the $6^{th}$ lines. Except that, FCL also introduces little PSNR and SSIM gains, showing its compatibility and potential benefits for image reconstruction. To further demonstrate the generalization of the FCL, we combine FCL with several state-of-the-art SR methods, as shown in Tab. 4. It could be seen that the FCL widely increases their performances, solidly demonstrating its generalization.

**Effect of FGG+FGC:** In the $3^{rd}$ line, we introduce the FG-Modules. Compared with the $1^{st}$ line, we could find that directly introducing the FG-Modules is trivial, even bringing large FE downgradation. Comparing $2^{nd}$ and $4^{th}$ lines could derive similar conclusions. It shows that more parameters are trivial. However, when we introduce our flux cues in the $5^{th}$ line, it brings a significant increase across all metrics. The 0.1+ FE decrease makes the model achieve high flux consistency without the FCL. Finally, when we introduce FCL, the performance has further gains, showing the compatibility of our different proposed modules.

### 5.4 Quality Analysis

Fig. 3 visually compares various super-resolution models on star field regions. The FISR model demonstrates superior reconstruction quality over traditional CNN-based methods such as EDSR and RCAN. Although these baselines can recover structures, they often fail to preserve flux consistency, particularly in regions with bright or overlapping celestial sources.

Table 3: Ablation study on the effectiveness of the Flux Guidance Generation (FGG) and Flux Error (FE). Metrics are reported on ×2 SR task.

|  | FG-Modules | Flux cues | FCL | PSNR↑ | SSIM↑ | FE↓ |
|---|---|---|---|---|---|---|
| $1^{st}$ |  |  |  | 37.7715 | 0.8288 | 0.7022 |
| $2^{nd}$ |  |  | ✓ | 37.8570 | 0.8283 | 0.6389 |
| $3^{rd}$ | ✓ |  |  | 37.7101 | 0.8286 | 0.7572 |
| $4^{th}$ | ✓ |  | ✓ | 37.8681 | 0.8301 | 0.7467 |
| $5^{th}$ | ✓ | ✓ |  | 37.8454 | 0.8302 | 0.6527 |
| $6^{th}$ | ✓ | ✓ | ✓ | 37.8779 | 0.8311 | 0.5739 |

Table 4: Comparison of different methods with and without FCL under ×2 and ×4 ASR.

| Scale | Flux Loss | EDSR [8] | RealESRGAN [34] | RCAN [32] | SwinIR [26] | PromptIR [27] | HAT [29] |
|---|---|---|---|---|---|---|---|
| ×2 | PSNR↑ (w/o) | 35.3816 | 36.8363 | 36.3703 | 37.2205 | 37.7715 | 37.2501 |
|  | SSIM↑ (w/o) | 0.8054 | **0.8225** | 0.8240 | **0.8286** | **0.8288** | 0.8295 |
|  | Flux Error↓ (w/o) | 1.4623 | 7.3632 | 0.9237 | 0.8130 | 0.7022 | 0.7636 |
|  | PSNR↑ (w/) | **35.6259** | **36.8647** | **37.6441** | 37.5098 | **37.8570** | **38.0880** |
|  | SSIM↑ (w/) | **0.8064** | 0.8222 | **0.8291** | 0.8280 | 0.8283 | **0.8320** |
|  | Flux Error↓ (w/) | **1.3334** | **6.4402** | **0.6631** | **0.7809** | **0.6389** | **0.6042** |
| ×4 | PSNR↑ (w/o) | 33.8736 | 34.3725 | **34.9823** | 34.6655 | 34.6726 | 34.9142 |
|  | SSIM↑ (w/o) | 0.7201 | 0.7223 | **0.7263** | 0.7258 | 0.7230 | 0.7276 |
|  | Flux Error↓ (w/o) | 1.3841 | 4.0782 | **1.0550** | 1.0657 | 1.0800 | 1.0256 |
|  | PSNR↑ (w/) | **34.2381** | **34.9243** | 34.2024 | **35.1610** | **35.0936** | **35.3156** |
|  | SSIM↑ (w/) | **0.7205** | **0.7234** | 0.7234 | **0.7265** | **0.7253** | **0.7280** |
|  | Flux Error↓ (w/) | **1.3659** | **1.1219** | 1.0755 | **1.0634** | **1.0227** | **1.0199** |

To further emphasize the impact of our flux consistency loss, we compute the Kullback-Leibler (KL) [70] and Jensen-Shannon (JS) [71] divergence between the predicted and ground-truth intensity distributions within selected regions. Notably, FISR—especially when trained with the flux loss—achieves significantly lower divergence scores, reflecting improved flux preservation and more physically accurate reconstructions.

## 5.5 Evaluation on Downstream Scientific Tasks

To address concerns about error propagation to scientific outputs, we evaluated our method on two representative downstream tasks: **stellar mass estimation** and **weak lensing shear measurement**.

For **stellar mass estimation**, we applied a simplified photometric pipeline to the STAR test set, converting predicted fluxes to magnitudes (mag $= -2.5 \times \log_{10}(\text{flux}) + 25.0$) and inferring stellar masses using a constant mass-to-light ratio ($M/L \approx 3.0$). As shown in Table 5, our FISR method achieves the lowest predicted magnitude error ($1.66 \times 10^{-7}$) alongside RealESRGAN ($1.67 \times 10^{-7}$),

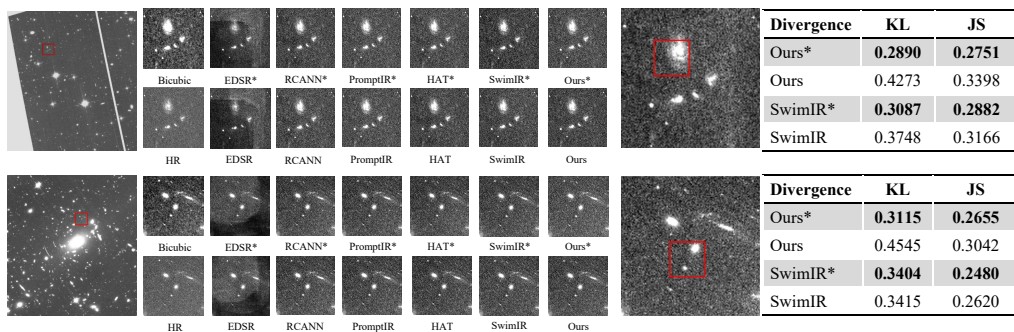

Figure 3: Visual comparison on two star field regions. Red boxes mark areas for computing KL and JS divergence between predictions and ground truth. Models with (∗) are trained using FCL.

Table 5: Evaluation on downstream scientific tasks. We report predicted magnitude error, stellar mass MAE, and mean shear error for different methods. Lower values indicate better performance.

| Method | Pred. Mag. Error | Mass MAE | Mean Shear Error |
|--------|------------------|----------|------------------|
| Bicubic | $3.23 \times 10^{-7}$ | $1.79 \times 10^{-7}$ | $2.10 \times 10^{-1}$ |
| SwinIR | $2.02 \times 10^{-7}$ | $5.19 \times 10^{-8}$ | $1.98 \times 10^{-1}$ |
| EDSR | $2.96 \times 10^{-7}$ | $1.37 \times 10^{-7}$ | $2.14 \times 10^{-1}$ |
| RCAN | $2.01 \times 10^{-7}$ | $5.36 \times 10^{-8}$ | $2.06 \times 10^{-1}$ |
| HAT | $1.67 \times 10^{-7}$ | $3.06 \times 10^{-8}$ | $1.88 \times 10^{-1}$ |
| RealESRGAN | $3.37 \times 10^{-7}$ | $3.99 \times 10^{-7}$ | $1.95 \times 10^{-1}$ |
| **FISR (Ours)** | $\mathbf{1.66 \times 10^{-7}}$ | $\mathbf{2.81 \times 10^{-8}}$ | $\mathbf{1.87 \times 10^{-1}}$ |

and demonstrates the best stellar mass MAE ($2.81 \times 10^{-8}$), significantly outperforming other methods including EDSR ($1.37 \times 10^{-7}$) and Bicubic ($1.79 \times 10^{-7}$).

For **weak lensing shear measurement**, a critical task for cosmological studies, we computed shear components $\gamma = \gamma_1 + i\gamma_2$ from SEP-detected galaxy ellipses, where:

$$\gamma_1 = \frac{a^2 - b^2}{a^2 + b^2} \times \cos(2\theta) \tag{6}$$

$$\gamma_2 = \frac{a^2 - b^2}{a^2 + b^2} \times \sin(2\theta) \tag{7}$$

with semi-major axis $a$, semi-minor axis $b$, and position angle $\theta$. We measured the mean shear difference $|\gamma_{\mathrm{pred}} - \gamma_{\mathrm{gt}}|$ between predictions and ground truth. The results in Table 5 show that FISR achieves competitive shear preservation ($1.87 \times 10^{-1}$), comparable to the best-performing HAT ($1.88 \times 10^{-1}$) and superior to methods like RCAN ($2.14 \times 10^{-1}$). These comprehensive evaluations demonstrate that our flux-preserving approach maintains both photometric accuracy and morphological fidelity essential for downstream scientific inference, validating its practical utility in astronomical applications.

## 6    Conclusion

In this work, we present STAR, a large-scale, flux-consistent benchmark specifically designed for astronomical super-resolution (ASR). Addressing critical limitations in prior datasets—such as flux inconsistency, object-centric bias, and limited diversity—STAR captures complex star fields with physically faithful flux distributions, cross-object interactions, and weak lensing effects. Alongside, we introduce a novel evaluation metric, Flux Error (FE), and propose the Flux-Invariant Super-Resolution (FISR) model, which incorporates flux-aware prompts and consistency loss. Extensive experiments show that FISR not only achieves state-of-the-art reconstruction quality but also significantly improves flux consistency.

## 7    Limitations and future work

While our study offers promising insights, it has a few limitations that merit further exploration. First, our experiments are based on observations from a single telescope, the HST WFC/ACS with the F814W filter, which may limit the generalizability of our findings to other instruments or observational contexts. Additionally, although our network design performs well, it could benefit from incorporating more domain-specific optimizations rooted in astronomical knowledge, such as leveraging physical principles or astronomical priors to enhance performance in complex scenarios. These areas present opportunities for future refinement. Looking forward, we aim to broaden the applicability of our method by extending it to a wider array of advanced telescopes, such as the James Webb Space Telescope (JWST) [72] or the upcoming Large Synoptic Survey Telescope (LSST) [73], to explore its potential across diverse astronomical contexts. Through these efforts, we hope to make modest contributions to the field of astronomical image processing, fostering the development of more robust and adaptable tools for future discoveries.

# 8 Acknowledgments

This work was supported by the Shanghai Artificial Intelligence Laboratory. This work was also supported by the JC STEM Lab of AI for Science and Engineering, funded by The Hong Kong Jockey Club Charities Trust, and the Research Grants Council of Hong Kong (Project No. CUHK14213224). We sincerely thank Hao Du, Yating Liu, Jiaze Li, Yingfan Hua, and Jun Yao for their invaluable guidance and insightful feedback throughout this research.

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

# Appendix

# A Flux Consistency Downsampling Details

**Computing image plane coordinates:** For a given high-resolution (HR) image with the resolution of $H \times W$ and a downsampling rate $s$, we generate the size of the downsampled low-resolution (LR) image, referred to as $\frac{H}{s} \times \frac{W}{s}$. With the two sizes, we have specific coordinates of pixels in both LR and HR images.

**Transfer pixels to sky:** For HR and IR pixel coordinates, we transfer them into the celestial coordinate system as: $(u, v) \rightarrow (ra, dec)$, where $(u, v)$ is a coordinate in the image plane while $(ra, dec)$ is the longitude and latitude coordinates of the Earth. Note that, each pixel is not an ideal point and actually a rectangle on the image plane. After the mapping, it becomes a quadrilateral surface of the celestial coordinate system. The physical meaning of this quadrilateral surface is the sky area covered by a pixel, denoted as the receptive field here. For the $i$-th pixel of the LR image and the $j$-th pixel of the HR image, we calculate and denote the area value of their receptive field as $A_i^{LR}$ and $A_j^{HR}$.

The transformation process in the aforementioned process is implemented by the telescope calibration information carried by the high-resolution (HR) image, which could be interpreted as camera intrinsic and extrinsic parameters of the telescope.

**Low-resolution image Flux Computation:** The previous steps essentially transferred HR and LR image plane grids into two surface meshes in the celestial coordinate system, as shown in Fig. 4. Obviously, the average receptive field of the LR image is larger than the HR one because the LR pixel corresponds to larger regions, leading to an LR pixel covers multiple HR pixels in the sky. To compute the flux of the $i$-th LR pixel, we first identify the set of HR pixels $S_i^o$ whose receptive fields overlap with that of the $i$-th LR pixel, i.e., $S_i^o = \{j \mid A_j^{HR} \cap A_i^{LR} \neq \emptyset\}$. This set represents all HR pixels whose sky areas contribute to the $i$-th LR pixel's flux. The flux of the $i$-th LR pixel, $F_i^{LR}$, is then computed by summing the weighted contributions from all overlapping HR pixels:

$$F_i^{LR} = \sum_{j \in S_i^o} w_{i,j} \cdot f_j^{HR}, \tag{8}$$

where $f_j^{HR}$ is the flux of the $j$-th HR pixel, and $w_{i,j}$ is the weight representing the fractional contribution of the $j$-th HR pixel to the $i$-th LR pixel.

The weight $w_{i,j}$ is calculated as:

$$w_{i,j} = \frac{A_{i,j}}{A_j^{HR}}, \tag{9}$$

where $A_{i,j}$ is the overlapping sky area between the $i$-th LR pixel and the $j$-th HR pixel, representing their shared quadrilateral patch in the celestial coordinate system, and $A_j^{HR}$ is the total sky area covered by the $j$-th HR pixel's receptive field. To better understand the role of this weight in flux computation, we substitute $w_{i,j}$ into Equation (8), transforming the contribution term as follows. The flux contribution from the $j$-th HR pixel to the $i$-th LR pixel is $w_{i,j} \cdot f_j^{HR}$, where $f_j^{HR}$ is the flux of the $j$-th HR pixel. Substituting $w_{i,j} = \frac{A_{i,j}}{A_j^{HR}}$ into this term, we obtain:

$$w_{i,j} \cdot f_j^{HR} = \left( \frac{A_{i,j}}{A_j^{HR}} \right) \cdot f_j^{HR} = A_{i,j} \cdot \frac{f_j^{HR}}{A_j^{HR}}. \tag{10}$$

Here, $\frac{f_j^{HR}}{A_j^{HR}}$ represents the flux density of the $j$-th HR pixel, i.e., the photon count per unit sky area, as recorded by the telescope's CCD sensor over the receptive field area $A_j^{HR}$. Thus, $A_{i,j} \cdot \frac{f_j^{HR}}{A_j^{HR}}$ is the flux contributed by the $j$-th HR pixel over the overlapping area $A_{i,j}$, ensuring that the contribution is proportional to the shared sky area between the LR and HR pixels. This approach preserves the total photon flux during downsampling, maintaining flux consistency across resolutions.

As shown in Fig. 5, we compare flux consistency downsampling with traditional bilinear interpolation. It can be found that the result of Fig. 5 (a) is closer to the average flux of HR star sources, indicating that flux consistency downsampling can better keep the original HR flux information. To further highlight the differences between the two methods, we visualize their residuals in Fig. 5 (c). Noticeable

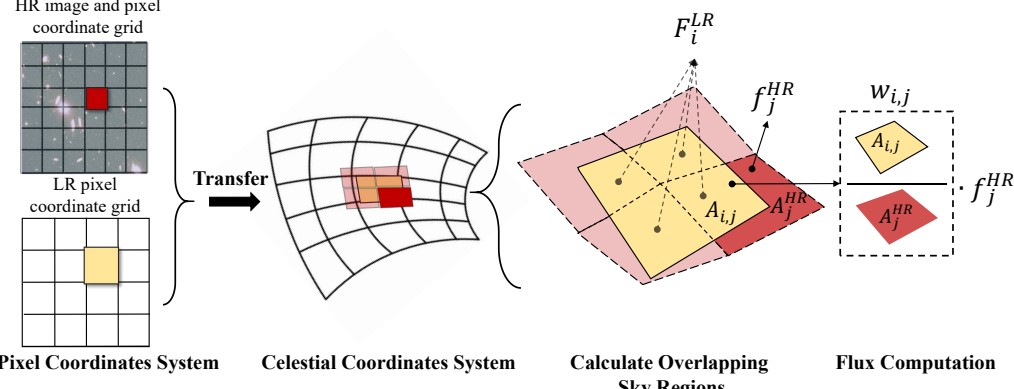

Figure 4: Schematic diagram of the flux-consistent downsampling process. The workflow illustrates the transformation of HR and LR image pixels into the celestial coordinate system, the computation of overlapping sky regions between HR and LR receptive fields, and the flux calculation for LR pixels using weighted contributions from overlapping HR pixels.

differences can be observed at the locations of stellar sources. The bilinear interpolation method tends to cause flux reduction when handling bright targets such as stars, making it less suitable for flux consistency astronomical applications.

## B  PSF Details

We simulate the imaging blur in the STAR dataset using two PSF models: the Gaussian PSF and the Airy PSF [74], aiming to increase training data diversity. The Gaussian PSF is a simple model often used to approximate blur in astronomical observations [75, 76]. In contrast, the Airy PSF captures diffraction effects from a telescope's circular aperture, making it suitable for space-based instruments like HST [24].

In the Gaussian PSF and Airy PSF models, $\sigma$ and $r$ serve as adjustable parameters to control the spread of the blur by modulating the energy dispersion of the filter. For instance, in the Gaussian PSF, a larger $\sigma$ leads to a less concentrated signal with greater energy spread across the filter, while in the Airy PSF, $r$ governs the radial extent of energy distribution due to diffraction, as defined below.

$$\text{PSF}_{\text{Gaussian}}(x, y) = \exp\left(-\frac{x^2 + y^2}{2\sigma^2}\right), \tag{11}$$

and

$$\text{PSF}_{\text{Airy}}(r) = \left[\frac{2J_1(kr)}{kr}\right]^2. \tag{12}$$

We define these parameters based on the telescope's observed blur characteristics, following Schawinski et al. [51], who used the observed blur to set the PSF parameters for a realistic simulation of hardware-specific degradation effects. Accordingly, we set the Gaussian PSF parameter $\sigma \in [\mathbf{0.8}, \mathbf{1.2}]$ and the Airy PSF radius $\mathbf{r} \in [\mathbf{1.9}, \mathbf{2.2}]$ pixels based on the FWHM [60], which measures the blur width at half its peak intensity, to approximate the actual HST WFC/ACS F814W filter observations where the blur is characterized by its FWHM. This enables effective super-resolution training.

## C  Additional Experiments with Gaussian + Airy PSFs

The original submission focuses on experiments using Gaussian PSF data. Here, we further evaluate the combination of Gaussian PSF and Airy PSF (Gaussian + Airy PSFs) and validate the effectiveness

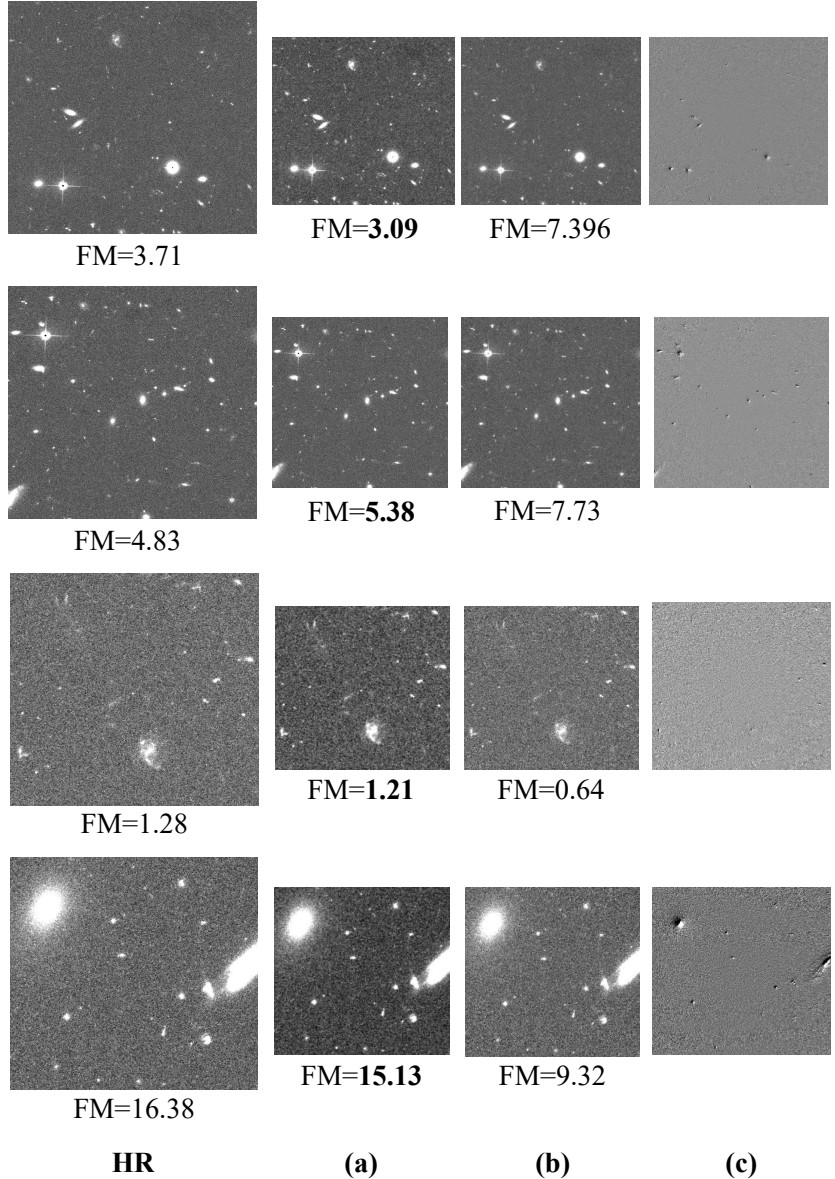

Figure 5: Comparison between downsampling methods. Each HR patch is shown alongside three columns: (a) flux-consistent downsampling, (b) bilinear interpolation, and (c) their pixel-wise difference. FM means flux mean.

Table 6: Performance of different methods under ×2 super-resolution with Gaussian PSF and Airy PSF on the STAR dataset. Metrics: PSNR↑, SSIM↑, Flux Error (FE)↓.

| Metric | Bicubic | EDSR | RCAN | SwinIR | FISR |
|--------|---------|--------|--------|--------|---------|
| PSNR | 29.4434 | 35.7398 | 37.4639 | 37.1347 | **38.2678** |
| SSIM | 0.7125 | 0.8086 | 0.8277 | 0.8279 | **0.8334** |
| FE | 4.286 | 1.3249 | 0.7451 | 0.7593 | **0.5585** |

Table 7: Performance of different methods under ×2 super-resolution with Gaussian PSF and Airy PSF on the STAR dataset (with and without FCL). Metrics: PSNR↑, SSIM↑, Flux Error (FE)↓.

| Flux Loss | Metric | EDSR | RCAN | SwinIR |
|-----------|--------|--------|--------|--------|
| w/o | PSNR | 35.7398 | 37.4639 | 37.1347 |
| | SSIM | 0.8086 | 0.8277 | 0.8279 |
| | FE | 1.3249 | 0.7451 | 0.7593 |
| w/ | PSNR | **35.8921** | **37.8914** | **37.6049** |
| | SSIM | **0.8092** | **0.8286** | **0.8281** |
| | FE | **1.242** | **0.5914** | **0.6767** |

of Flux-Consistent Loss (FCL) in this setting. In this setting, each image is degraded by randomly selecting either the Gaussian or Airy PSF with equal probability.

We compare the performance of different methods under ×2 super-resolution with Gaussian PSF and Airy PSF on the STAR dataset, analyzing the results model-wise and loss-wise. Tab. 6 compares the performance of all methods in this setting. FIRS surpasses baselines like SwinIR and RCAN, achieving a 3.05% higher PSNR and 26.45% lower FE than SwinIR, demonstrating its superior ability to recover fine stellar details and preserve flux accuracy in astronomical image super-resolution. Additionally, Tab. 7 compares EDSR, RCAN, and SwinIR with and without FCL to focus on the impact of FCL across baseline methods. For instance, SwinIR with FCL improves PSNR by 1.27% and reduces FE by 10.88% compared to the version without FCL, while RCAN with FCL improves PSNR by 1.14% and reduces FE by 20.63%, highlighting FCL's role in enhancing image quality and flux preservation.

## D   Additional visualizations

We present additional visualizations to demonstrate the effectiveness of our approach in star-field super-resolution (ASR) tasks. Fig. 6 displays the ×2 super-resolution results for the Gaussian PSF experiment, comparing baselines (EDSR, RCAN, PromptIR, SwinIR, HAT) against our FIRS model. The visualizations reveal that FIRS consistently outperforms all baselines, achieving superior visual quality with finer stellar details and sharper structures. To further quantify these improvements, we compute the KL divergence and JS divergence between the intensity distributions of the predicted and ground truth values in selected regions, following the experimental settings in the original submission. The results show that FIRS significantly reduces distribution discrepancies compared to SwinIR and HAT, confirming its superior capability in preserving stellar details and flux accuracy in ASR tasks.

## E   Hyperparameters Tuning

We tune the parameter $\lambda$ to balance the Flux-Consistent Loss (FCL) and reconstruction loss in our star-field super-resolution (ASR) model. We evaluate different $\lambda$ values (0.1, 0.05, and 0.01) under the ×2 Gaussian PSF + Airy PSF setting, with results shown in Tab. 17. The performance metrics show that $\lambda = 0.01$ yields the best results, improving PSNR by 1.40% and reducing FE by 15.88% compared to $\lambda = 0.1$. These results indicate that a proper $\lambda$ matters in the balance between reconstruction loss and the Flux-Consistent Loss. Fortunately, 0.01 seems to perform well in most cases.

## F  Experimental setting/details

We ensure reproducibility by providing the experimental environment and computational resources. Tab. 8 shows the environment configuration, including hardware and software details. Tab. 9 summarizes the computational resources used for training. For detailed training settings and parameters of each model, please see the code.

Table 8: Experimental Environment Setup.

| Component | Version |
|---|---|
| OS | Ubuntu 20.04.5 LTS |
| Python | 3.10.15 |
| PyTorch | 2.0.0 |
| CUDA | 11.8 |

Table 9: Computational Resources for Different Methods (Training Time in Hours).

| Method | Training Time (Hours) |
|---|---|
| EDSR | 52 |
| RCAN | 40 |
| Hat | 70 |
| SwinIR(light weight) | 14 |
| PromptIR | 15 |
| GAN | 27 |
| FISR (ours) | 15 |

## G  Additional Experiments

To further validate the robustness and scientific utility of our proposed dataset and model, we conducted a series of additional experiments in response to reviewer feedback. These experiments evaluate the model's generalization capabilities across different domains, its performance on downstream scientific tasks, its robustness to noise, and its computational efficiency.

### G.1  Generalization and Robustness Analysis

**Cross-Filter Generalization:** To test the model's performance on data from different instrumental filters, we evaluated our F814W-trained model on test sets from the F606w and F435w filters of the Hubble Space Telescope (HST). As shown in Tab. 10, while there is a performance drop as the filter domain shifts further from the training domain (F814W), the model maintains strong performance, demonstrating satisfactory generalization capabilities. The F606w filter, being spectrally closer to F814W, yields better results than the more distant F435w filter, confirming that domain similarity influences performance.

Table 10: Cross-filter generalization performance of the FISR model trained on the F814W filter.

| Metric | F435w | F606w | F814w (In-Domain) |
|---|---|---|---|
| PSNR | 35.9192 | 36.3522 | 37.8779 |
| SSIM | 0.7305 | 0.7667 | 0.8311 |
| Flux Error | 0.9193 | 0.8242 | 0.5739 |

**Robustness to Noise:** We evaluated FISR's robustness by introducing random Poisson noise to each image during inference, simulating realistic observational noise. The results in Tab. 11 show that FISR maintains its state-of-the-art performance, achieving the best results across all metrics compared to other methods under noisy conditions.

**Cross-Dataset Evaluation:** Although direct evaluation is challenging due to differences in data units (STAR uses scientific counts, while AstroSR uses RGB), we re-trained our FISR model on the

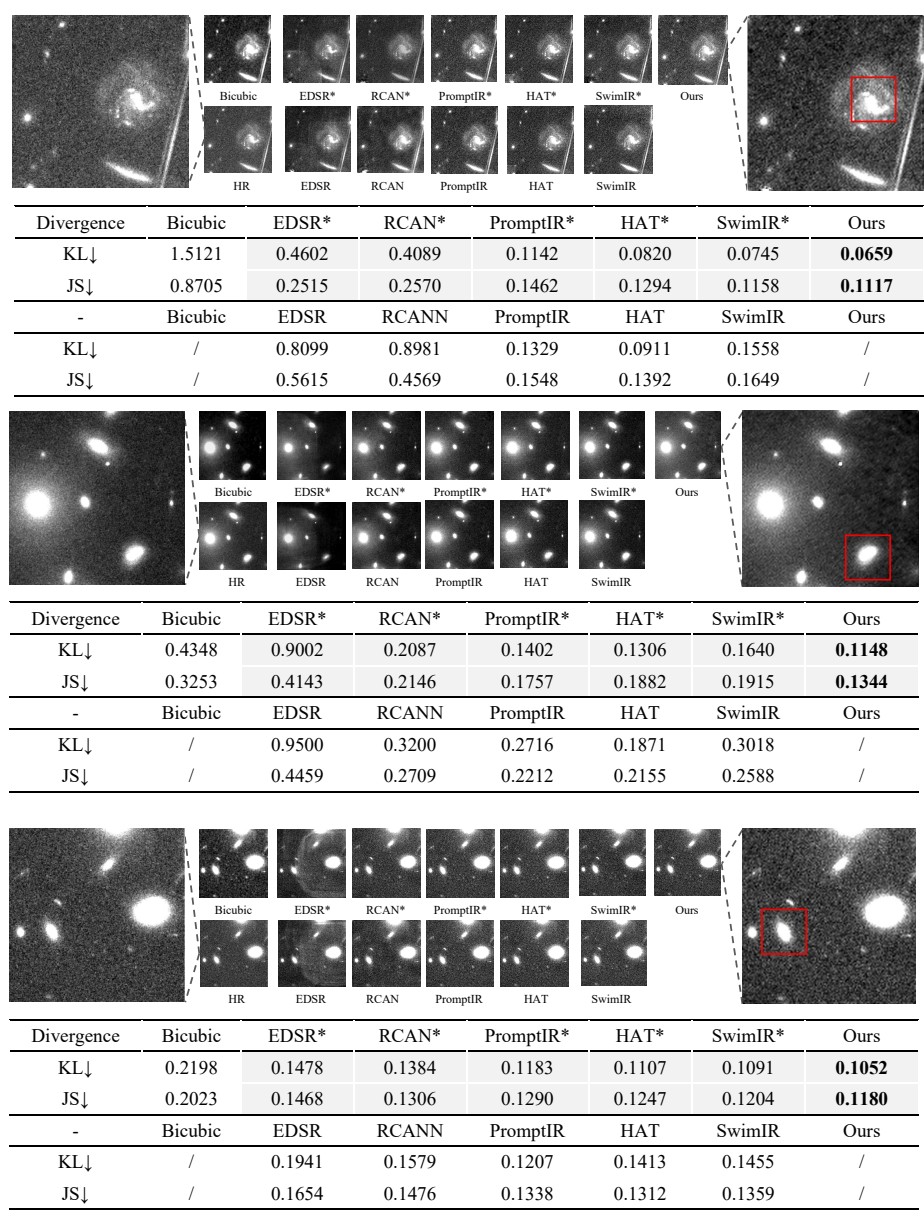

| Divergence | Bicubic | EDSR* | RCAN* | PromptIR* | HAT* | SwimIR* | Ours |
|---|---|---|---|---|---|---|---|
| KL↓ | 1.5121 | 0.4602 | 0.4089 | 0.1142 | 0.0820 | 0.0745 | **0.0659** |
| JS↓ | 0.8705 | 0.2515 | 0.2570 | 0.1462 | 0.1294 | 0.1158 | **0.1117** |
| - | Bicubic | EDSR | RCANN | PromptIR | HAT | SwimIR | Ours |
| KL↓ | / | 0.8099 | 0.8981 | 0.1329 | 0.0911 | 0.1558 | / |
| JS↓ | / | 0.5615 | 0.4569 | 0.1548 | 0.1392 | 0.1649 | / |

| Divergence | Bicubic | EDSR* | RCAN* | PromptIR* | HAT* | SwimIR* | Ours |
|---|---|---|---|---|---|---|---|
| KL↓ | 0.4348 | 0.9002 | 0.2087 | 0.1402 | 0.1306 | 0.1640 | **0.1148** |
| JS↓ | 0.3253 | 0.4143 | 0.2146 | 0.1757 | 0.1882 | 0.1915 | **0.1344** |
| - | Bicubic | EDSR | RCANN | PromptIR | HAT | SwimIR | Ours |
| KL↓ | / | 0.9500 | 0.3200 | 0.2716 | 0.1871 | 0.3018 | / |
| JS↓ | / | 0.4459 | 0.2709 | 0.2212 | 0.2155 | 0.2588 | / |

| Divergence | Bicubic | EDSR* | RCAN* | PromptIR* | HAT* | SwimIR* | Ours |
|---|---|---|---|---|---|---|---|
| KL↓ | 0.2198 | 0.1478 | 0.1384 | 0.1183 | 0.1107 | 0.1091 | **0.1052** |
| JS↓ | 0.2023 | 0.1468 | 0.1306 | 0.1290 | 0.1247 | 0.1204 | **0.1180** |
| - | Bicubic | EDSR | RCANN | PromptIR | HAT | SwimIR | Ours |
| KL↓ | / | 0.1941 | 0.1579 | 0.1207 | 0.1413 | 0.1455 | / |
| JS↓ | / | 0.1654 | 0.1476 | 0.1338 | 0.1312 | 0.1359 | / |

Figure 6: We further demonstrate several sets of visualization results on the ×2 Gaussian PSF experiment. Models with (*) are trained using FCL.

Table 11: Performance comparison under Poisson noise injection during inference. Best results are in **bold**.

| Method | Bicubic | EDSR | SwinIR | RCAN | HAT | RealESRGAN | FISR (Ours) |
|---|---|---|---|---|---|---|---|
| PSNR | 28.9823 | 34.8191 | 36.3945 | 35.2918 | 36.8342 | 35.8098 | **36.7803** |
| SSIM | 0.6825 | 0.7684 | 0.7883 | 0.7848 | 0.7743 | 0.7852 | **0.7888** |
| Flux Error | 4.7889 | 1.5682 | 1.1433 | 1.1993 | 1.1943 | 6.9292 | **1.1025** |

AstroSR dataset. Tab. 12 demonstrates that our method outperforms the original baseline models reported in the AstroSR paper, showcasing its architectural effectiveness on different data types.

Table 12: Performance comparison on the AstroSR dataset after re-training. Best results are in **bold**.

| Method | Bicubic | EDSR | RCAN | ENLCA | SRGAN | FISR (Ours) |
|---|---|---|---|---|---|---|
| PSNR | 17.7714 | 23.2168 | 23.6082 | 23.4267 | 23.0039 | **24.0211** |
| SSIM | 0.1686 | 0.3910 | 0.3966 | 0.3963 | 0.3854 | **0.4025** |
| Flux Error | 233.2564 | 50.5872 | 61.3863 | 59.1659 | 42.3078 | **33.2331** |

## G.2 Evaluation on Downstream Scientific Tasks

To quantify the practical impact of our super-resolution model on real-world scientific analysis, we evaluated its performance on four representative downstream astronomical tasks. These experiments are designed to demonstrate that improvements in standard metrics like PSNR, SSIM, and our proposed Flux Error (FE) directly translate to higher fidelity in scientific measurements. The methodologies and results for these tasks are detailed below, with a final comparative summary in Table 13.

## G.3 Evaluation on Downstream Scientific Tasks

To quantify the practical impact of our super-resolution model on real-world scientific analysis, we evaluated its performance on two representative downstream astronomical tasks. These experiments are designed to demonstrate that improvements in standard metrics and our proposed Flux Error (FE) directly translate to higher fidelity in scientific measurements. The methodologies and results for these tasks are detailed below.

**Object Detection Sensitivity:** The ability to detect faint objects is fundamental to astronomical surveys, determining the depth and completeness of celestial catalogs. An effective SR model should enhance faint sources, thereby improving detection sensitivity. In our experiment, we performed bipartite matching between sources detected in the predicted images and the ground-truth catalog, with a match considered successful if the spatial distance was within 2 pixels. The sensitivity was quantified using the **Recall** metric. Our FISR model achieves a high recall of **81.47%**, indicating strong performance in identifying celestial objects.

**Distance Estimation:** Accurately measuring the distances to celestial objects is a cornerstone of cosmology, combining both object detection and precise photometry. To evaluate this, we used the successfully matched object pairs from the detection task. We converted each object's flux to an apparent magnitude ($m$) and then applied the distance modulus formula, $d = 10^{(m-M+5)/5}$, to estimate the distance ($d$) in megaparsecs (MPC), assuming a constant absolute magnitude ($M$) of 4.83 (typical for Sun-like stars). The accuracy was evaluated by the Mean Absolute Error (MAE) between the predicted and ground-truth distances, with the results shown in Table 13.

Table 13: Evaluation on the downstream task of distance estimation. Lower values indicate better performance. Best results are in **bold**.

| Metric | Bicubic | SwinIR | EDSR | RCAN | HAT | R-ESRGAN | FISR |
|---|---|---|---|---|---|---|---|
| Distance MAE (MPC) | 6.82E+03 | 5.37E+03 | 6.44E+03 | 5.61E+03 | 4.44E+03 | 4.89E+03 | **4.12E+03** |

## H More Ablation Studies on FGG Module

We performed ablation studies to analyze the sensitivity of the Flux Guidance Generation (FGG) module.

**Kernel Choice in FGG:** We tested alternative kernels (Airy, and a random mix of Gaussian/Airy) for rendering the flux map. Tab. 14 shows that performance remains stable across different kernel choices, suggesting that the module's primary function is to provide a spatial prior for flux information, rather than depending on a specific kernel formulation.

Table 14: Ablation study on the kernel choice within the FGG module.

| Kernel Type | PSNR | SSIM | Flux Error |
|---|---|---|---|
| Gaussian | 37.8779 | 0.8311 | 0.5739 |
| Airy | 37.6988 | 0.8305 | 0.5664 |
| Gaussian/Airy (Random) | 37.8186 | 0.8311 | 0.5726 |

**Sensitivity to Detection Errors:** To assess FGG's robustness, we introduced noisy detections by lowering the source detection threshold, resulting in twice the number of sources, including many false positives. As seen in Tab. 15, while performance degrades slightly, FISR remains robust and achieves results comparable to the model trained with clean detections. This indicates that the model's performance does not solely depend on the precision of the FGG's input.

Table 15: Performance of FISR with clean versus noisy source detections in the FGG module.

| Detection Quality | PSNR | SSIM | Flux Error |
|---|---|---|---|
| Clean Detections | 37.8779 | 0.8311 | 0.5739 |
| Noisy Detections | 37.3176 | 0.8275 | 0.6872 |

## I Computational Efficiency

We measured the single-image inference time for all compared methods. The results in Tab. 16 show that FISR is computationally efficient, with an inference time comparable to other high-performing transformer-based models like SwinIR.

Table 16: Inference time per image (in seconds) for various SR methods.

| Method | Bicubic | EDSR | SwinIR | RCAN | HAT | RealESRGAN | FISR (Ours) |
|---|---|---|---|---|---|---|---|
| Time (s) | 0.0014 | 0.1908 | 0.1088 | 0.1237 | 0.6747 | 0.0995 | 0.1698 |

## J Limitations and future work

While our study offers promising insights, it has a few limitations that merit further exploration. First, our experiments are based on observations from a single telescope, the HST WFC/ACS with the F814W filter, which may limit the generalizability of our findings to other instruments or observational contexts. Additionally, although our network design performs well, it could benefit from incorporating more domain-specific optimizations rooted in astronomical knowledge, such as leveraging physical principles or astronomical priors to enhance performance in complex scenarios. These areas present opportunities for future refinement. Looking forward, we aim to broaden the applicability of our method by extending it to a wider array of advanced telescopes, such as the James Webb Space Telescope (JWST) [72] or the upcoming Large Synoptic Survey Telescope (LSST) [73], to explore its potential across diverse astronomical contexts. Furthermore, we plan to enhance our network design by integrating more astronomy-driven optimizations, incorporating physical knowledge and astronomical priors to better address challenges like crowded stellar regions or variable noise conditions. Through these efforts, we hope to make modest contributions to the

field of astronomical image processing, fostering the development of more robust and adaptable tools for future discoveries.

Table 17: Ablation study on the penalty factor $\lambda$ ($\times 2$ on Gaussian PSF + Airy PSF).

| FCL Weight $\lambda$ | PSNR↑ | SSIM↑ | FE↓ |
|---|---|---|---|
| 0.1 | 37.0843 | 0.8198 | 0.7842 |
| 0.05 | 37.2672 | 0.8252 | 0.7064 |
| 0.01 | 37.6049 | 0.8281 | 0.6767 |

## K  More Visualizations of the STAR Dataset

To further illustrate the unique characteristics and scale of the STAR benchmark, this section provides additional visualizations of the source data. We present examples of the original, full-frame observational images from the Hubble Space Telescope (HST) WFC/ACS instrument, which constitute the raw data prior to the patch subdivision process for model training 7.

These wide-field views underscore a core advantage of STAR over previous object-centric datasets. Instead of focusing on isolated, cropped targets, our dataset provides a holistic view of extensive celestial regions, preserving the crucial spatial context and inter-object relationships (e.g., cross-object interaction, weak lensing). Furthermore, we showcase a gallery of selected image patches to highlight the rich diversity within STAR 8. These examples span a wide range of astronomical environments, from dense, crowded stellar fields and sparsely populated regions to complex nebulae and fields containing multiple galaxies. Collectively, these visualizations reinforce the value of STAR as a comprehensive and physically faithful benchmark for advancing astronomical super-resolution research.

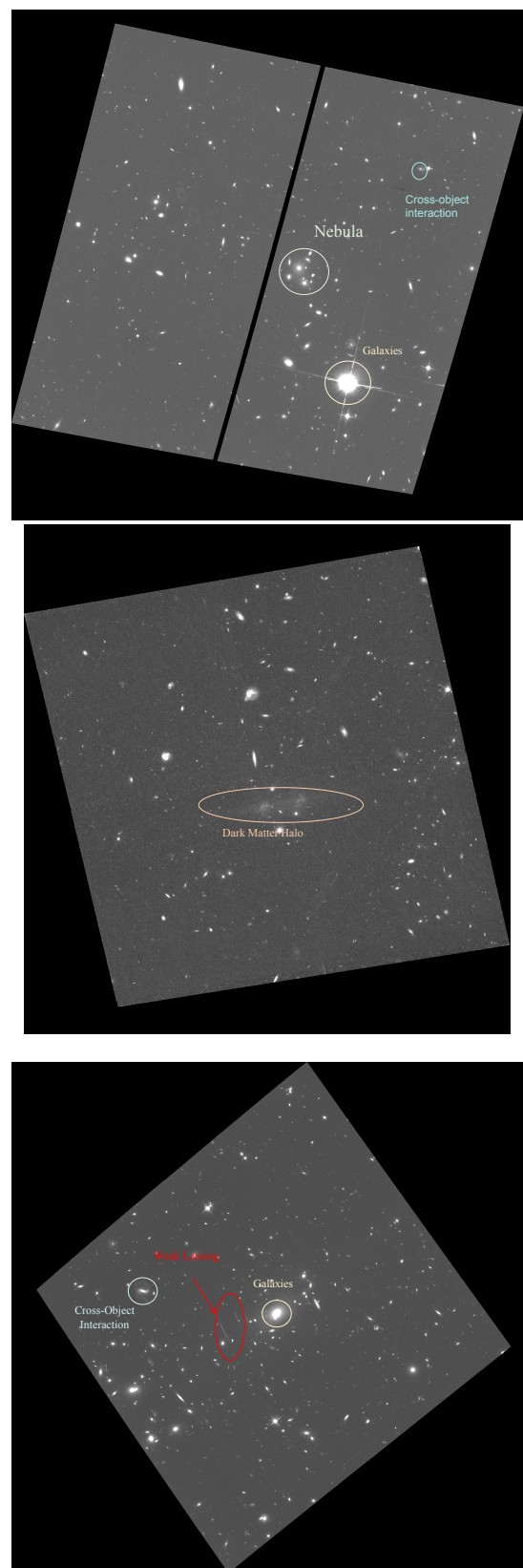

Figure 7: Examples of the original wide-field raw data from the HST WFC/ACS survey, which form the basis of the STAR dataset.

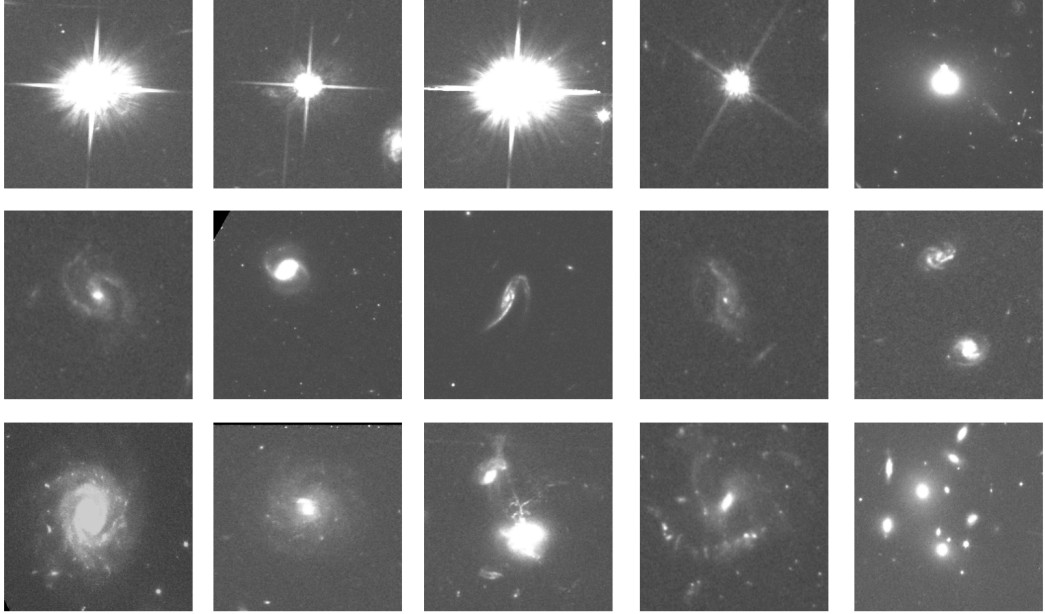

Figure 8: A selection of patches from the STAR dataset, showcasing its diversity. The examples include crowded stellar fields, regions with interacting galaxies, and complex nebulae, demonstrating the variety of astronomical scenes available for training robust models.

