# OpenReview forum: "STAR: A Benchmark for Astronomical Star Fields Super-Resolution"
_NeurIPS.cc/2025/Datasets_and_Benchmarks_Track — NeurIPS 2025 Datasets and Benchmarks Track spotlight_

### Official Review · Reviewer_guC8 · 2025-06-06

**Rating:** 4
**Confidence:** 3

**Summary:**

This paper proposes STAR, a new large-scale dataset for astronomical super-resolution (ASR) that addresses major shortcomings in existing datasets. Prior datasets suffer from flux inconsistency, object-centric design, and limited diversity. STAR contains 50,000+ flux-consistent image pairs derived from Hubble Space Telescope data. The authors introduce Flux Error, a metric that measures the discrepancy in total light per object between super-resolved and ground-truth images. Additionally, they propose the FISR model, which incorporates flux-aware design and training objectives. Experimental results demonstrate that FISR achieves superior performance over existing methods. The dataset and code are publicly released.

**Dataset Code Accessibility:**

Yes

**Ethical Considerations:**

No, there are no or only very minor ethics concerns

**Final Justification:**

My concerns have been addressed. I will maintain my rating.

**Limitations Weaknesses:**

1) SR methods for stars should emphasize photometric fidelity, while for galaxies and nebulae, morphological and structural accuracy becomes equally or more important. The paper provides limited discussion on the generalizability of STAR or FISR to non-stellar domains such as galaxies or nebulae.
2) The visualization seems not distinctive enough to show the superority of the mehotd compared with others; However, I believe that human visual comparison is less important as the flux error which may affect scientific analze. However, the paper's evaluation seems to be short of directly linking flux error to downstream astronomical tasks, such as object detection sensitivity, or distance estimation. Including such task-level evaluations would further strengthen the claim.
3) Is it possible to include cross-dataset experiments—such as training and evaluating the same SR model on both STAR and prior datasets (e.g., AstroSR, DiffLense)—to quantitatively validate STAR’s advantages in terms of model performance, physical fidelity, or generalization. This would strengthen the claim that STAR provides a more effective and realistic benchmark for astronomical super-resolution.
4) minor typo: L246: “They we use this map to weighted the pixel wised supervision ...”

**Strengths Contributions:**

The strengths of the paper include the introduction of a physically faithful and large-scale ASR dataset that captures field-level complexity, offering broader scientific utility. The proposed Flux Error metric is well-motivated for astrophysical applications and aligns model evaluation with scientific needs. The FISR model integrates domain knowledge about flux into both architecture and optimization, and shows consistent performance gains over baselines. The experimental section is overall thorough, including ablations and comparisons across multiple methods and settings. The authors also ensure reproducibility through public release of their code and dataset.

---

> ### Author Rebuttal · Authors · 2025-07-31
>
> Thanks for your valuable feedback and efforts.
>
> **1. SR methods for stars should emphasize photometric fidelity, while for galaxies and nebulae, morphological and structural accuracy becomes equally or more important. The paper provides limited discussion on the generalizability of STAR or FISR to non-stellar domains such as galaxies or nebulae.**
>
> Our dataset includes the largest possible collection of historical general survey data from the HST. The HST data covers a large number of galaxies and nebulae. We will add visualizations of the relevant images in the final revision.
>
> **2. The visualization seems not distinctive enough to show the superority of the mehotd compared with others; However, I believe that human visual comparison is less important as the flux error which may affect scientific analze. However, the paper's evaluation seems to be short of directly linking flux error to downstream astronomical tasks, such as object detection sensitivity, or distance estimation. Including such task-level evaluations would further strengthen the claim.**
>
> **Object Detection Sensitivity**: To better support the connection between flux error and downstream tasks, we have added evaluations on object detection sensitivity and distance estimation. For object detection sensitivity, we performed bipartite matching between predicted and ground-truth sources. A match is considered valid if the spatial distance is within 2 pixels, and the matched pairs are then used to compute recall as the sensitivity metric. Our method achieves a recall of 81.47%, indicating strong detection performance.
>
> **Distance Estimation**: To further enhance our analysis and validate the impact on distance estimation, we conducted an additional experiment to measure the distance from celestial objects to the observer. Specifically, we utilized the successfully matched pred and gt pairs from the previous bipartite matching, converting their respective flux values to apparent magnitudes using the formula m=−2.5log⁡10(flux)+zero_point. Then applied the distance modulus formula d=10(m−M+5)/5. with an assumed absolute magnitude M=4.83 (typical for sun-like stars) to estimate distances in megaparsecs (MPC). The evaluation metric chosen was MAE (Mean Absolute Error) between the predicted and ground-truth distances, reflecting the average deviation in distance estimation across all matched objects. The results are presented in the following table:
>
> |       | Bicubic   | SwinIR    | EDSR      | RCAN      | HAT       | RealESRGAN | ours(FISR) |
> |-------|-----------|-----------|-----------|-----------|-----------|------------|------------|
> | MPC | 6.82E+03 | 5.37E+03 | 6.44E+03 | 5.61E+03 | 4.44E+03 | 4.89E+03   | 4.12E+03   |
>
> **3. Is it possible to include cross-dataset experiments—such as training and evaluating the same SR model on both STAR and prior datasets (e.g., AstroSR, DiffLense)—to quantitatively validate STAR’s advantages in terms of model performance, physical fidelity, or generalization. This would strengthen the claim that STAR provides a more effective and realistic benchmark for astronomical super-resolution.**
>
> We thank the reviewer for this suggestion. Direct cross-dataset evaluation is not possible. AstroSR provides RGB images, while our dataset consists of HST scientific images (measured in counts, representing the number of photons per pixel), which are more suitable for astronomical research. In our knowledge, our dataset is only one star-field dataset whose data unit is the original scientific one.  We re-training our model on the AsrtoSR and adjusting the number of channels to accept RGB input. The results are as follows:
>
> |       | Bicubic  | EDSR    | RCAN    | ENLCA   | SRGAN   | FISR|
> |-------|----------|---------|---------|---------|---------|---------|
> | PSNR  | 17.7714  | 23.2168 | 23.608  | 23.4267 | 23.0039 | 24.0211 |
> | SSIM  | 0.1686   | 0.391   | 0.3966  | 0.3963  | 0.3854  | 0.4025  |
> | Flux Score | 233.2564 | 50.5872 | 61.3863 | 59.1659 | 42.3078 | 33.2331 |
>
> Bicubic, EDSR, RCAN, ENLCA, and SRGAN are all experimental methods in AstroSR. Since the code of the AstroSR is not open source, we reproduce and optimize the methods following the AstroSR paper and compare them with our method. The results show that our method still outperforms these methods. Since DiffLense does not have open source data and code, we are unable to conduct an effective evaluation on it due to the rebuttal limited time.
>
> **4. minor typo: L246: “They we use this map to weighted the pixel wised supervision ...**
>
> Thank you for noting the typo. We will fix it in the final revisions to ensure clarity.

---

### Official Review · Reviewer_vUSZ · 2025-06-16

**Rating:** 5
**Confidence:** 4

**Summary:**

This paper introduces ​STAR, a large-scale astronomical super-resolution dataset addressing critical limitations of existing datasets: flux inconsistency, object-centric cropping, and limited diversity. STAR provides 54,738 flux-consistent image pairs from Hubble Space Telescope data, covering complex star fields with cross-object interactions and weak lensing effects. The authors also propose Flux Error (FE)​, a physics-driven metric evaluating flux preservation in SR outputs, and ​Flux-Invariant Super-Resolution (FISR)​, a transformer-based model integrating flux-aware prompts via Flux Guidance Generation (FGG) and Flux Guidance Controller (FGC) modules, optimized with a Flux Consistency Loss (FCL).

**Additional Feedback:**

Here are some suggestive comments for authors:
Test cross-dataset generalization (e.g., AstroSR) and computational efficiency will improve the rigorous evaluation.
Formalize flux consistency as an optimization constraint (e.g., spectral preservation loss).
Analyze FGG’s robustness to detection noise/errors.

**Dataset Code Accessibility:**

Yes

**Ethical Considerations:**

No, there are no or only very minor ethics concerns

**Final Justification:**

The authors have well addressed my concerns, and I am willing to raise my score to accept.

**Limitations Weaknesses:**

1. Limited technical novelty of FISR architecture. FISR’s core encoder/decoder mirrors PromptIR, with FGG/FGC as add-on modules. While flux injection is novel, the base architecture lacks innovation.
2. FGG’s Gaussian-kernel rendering of object flux (Sec 4.2) is empirically designed. Justification for kernel choices (e.g., σ vs. ellipse size) or sensitivity to detection errors is absent.
3.  Claims of "flux propagation" lack formal analysis (e.g., how FCL ensures ∫(HR) dΩ ≈ ∫(LR) dΩ at object level).
4. The cross-dataset evaluation is missing. Tests are confined to STAR. Validation on existing ASR datasets (e.g., AstroSR) would demonstrate generalisation.
5. Flux preservation is motivated for photometry, but error propagation to scientific outputs (e.g., stellar mass estimates, weak lensing shear) is unquantified. It would be good to evaluate on downsteaming tasks.
6.  Sec 3.2 mentions Gaussian/Airy kernels but omits parameters (e.g., FWHM). Code release must clarify if PSFs are simulated or telescope-calibrated.

**Strengths Contributions:**

1. High-Impact dataset with rigorous physical faithfulness. The downsampling method emulates telescope CCD physics (§3.3), ensuring total flux preservation across resolutions. This corrects a critical flaw in existing datasets (e.g., AstroSR, DiffLense) that use naïve interpolation, which violates astronomical principles.
2. Meaningful physics-driven contributions, including FE metric, FCL loss, and FGG/FGC modules. FISR achieves ​SOTA FE/PSNR/SSIM​ across scales (×2/×4) under Gaussian+Airy blur

---

> ### Author Rebuttal · Authors · 2025-07-31
>
> Thanks for your valuable feedback and efforts.
>
> **1. Limited technical novelty of FISR architecture. FISR’s core encoder/decoder mirrors PromptIR, with FGG/FGC as add-on modules. While flux injection is novel, the base architecture lacks innovation.**
>
> The current model is our initial exploration of this task to serve as a validation of our dataset and a baseline for the community. The core design is to create a simple yet effective strong baseline. Therefore, we prioritized several features: (1) simple structure but highlight the task prior. (2) easy to secondary development. Based on this positioning, we did not consider designing more complex modules at this stage. We agree with your suggestion and will try to design more modules in future work.
>
> **2. FGG’s Gaussian-kernel rendering of object flux (Sec 4.2) is empirically designed. Justification for kernel choices (e.g., σ vs. ellipse size) or sensitivity to detection errors is absent.**
>
> 1. We added two experiments: one using the Airy kernel and the other combining a Gaussian kernel and the Airy kernel under random conditions. The experimental results are as follows:
>
> |       | Gaussian | Airy    | Gaussian/Airy |
> |-------|----------|---------|---------------|
> | PSNR  | 37.8779 | 37.6988 | 37.8186       |
> | SSIM  | 0.8311 | 0.8305  | 0.8311        |
> | flux score | 0.5739 | 0.5664  | 0.5726        |
>
> The results show that the specific form of the FGG kernel does not significantly affect the performance. We believe the kernel's goal is to incorporate flux information, primarily providing spatial prior information.
>
> 2. To evaluate the sensitivity of FGG to detection errors, we retrained a new model under the scenarios of deliberately introducing wrong detections. We lower the detection threshold, causing a 2-fold number of sources, including a large number of fake sources. The results are as follows:
>
> |       | Random Bad Det | FISR   |
> |-------|----------------|--------|
> | PSNR  | 37.3176        | 37.8779|
> | SSIM  | 0.8275         | 0.8311 |
> | flux score | 0.6872      | 0.5739 |
>
> Although the presence of noisy detections slightly degrades performance, FISR remains robust, achieving comparable PSNR, SSIM, and flux preservation with the clean one. This suggests that the overall performance of the model does not solely rely on the detection quality of FGG.
>
> **3. Claims of "flux propagation" lack formal analysis (e.g., how FCL ensures ∫(HR) dΩ ≈ ∫(LR) dΩ at object level).**
>
> We first clarify that the FCL enforces consistency in total flux between the high-resolution image and the predicted image, i.e.,∫ HR dΩ ≈ ∫ Prediction dΩ. Furthermore, the consistency between high-resolution and low-resolution images,  ∫ HR dΩ ≈ ∫ LR dΩ, is ensured by the design of our proposed dataset. The "flux propagate" we mentioned in line 67 refers to the process of flux propagating from the input to the output of the neural network. Currently, deep learning methods have difficulty establishing formal mathematical constraints for this process, and its effectiveness relies more on empirical loss function design.
>
> **4. The cross-dataset evaluation is missing. Tests are confined to STAR. Validation on existing ASR datasets (e.g., AstroSR) would demonstrate generalisation.**
>
> We thank the reviewer for this suggestion. Direct cross-dataset evaluation is not possible. AstroSR provides RGB images, while our dataset consists of HST scientific images (measured in counts, representing the number of photons per pixel), which are more suitable for astronomical research. In our knowledge, our dataset is only one star-field dataset whose data unit is the original scientific one. Therefore, our model cannot be easily operated the cross-dataset evaluation. So, we propose two alternative experiments.
>
> 1. We re-training our model on the AsrtoSR and adjusting the number of channels to accept RGB input. The results are as follows:
>
> |       | Bicubic  | EDSR    | RCAN    | ENLCA   | SRGAN   | FISR|
> |-------|----------|---------|---------|---------|---------|---------|
> | PSNR  | 17.7714  | 23.2168 | 23.608  | 23.4267 | 23.0039 | 24.0211 |
> | SSIM  | 0.1686   | 0.391   | 0.3966  | 0.3963  | 0.3854  | 0.4025  |
> | Flux Score | 233.2564 | 50.5872 | 61.3863 | 59.1659 | 42.3078 | 33.2331 |
>
> Bicubic, EDSR, RCAN, ENLCA, and SRGAN are all experimental methods in AstroSR. Since the code of the AstroSR is not open source, we reproduce and optimize the methods following the AstroSR paper and compare them with our method. The results show that our method still outperforms these methods.
>
> 2. We also performed a cross-filter evaluation. The other filter bands corresponding to F814w in HST are F606w and F435w. We selected the same amount of F606w and F435w data from the HST dataset as the test set and tested them using the model trained on F814w. The results are as follows:
>
> |       | F435w   | F606w   | F814w   |
> |-------|---------|---------|---------|
> | PSNR  | 35.9192 | 36.3522 | 37.8779 |
> | SSIM  | 0.7305  | 0.7667  | 0.8311  |
> | Flux Score | 0.9193 | 0.8242 | 0.5739 |
>
> These results demonstrate that our model also generalizes well to cross-filter performance.
>
> **5. Flux preservation is motivated for photometry, but error propagation to scientific outputs (e.g., stellar mass estimates, weak lensing shear) is unquantified. It would be good to evaluate on downsteaming tasks.**
>
> To address this concern, we conducted supplementary experiments on two representative use cases: stellar mass estimation and weak lensing shear measurement.
>
> 1. **Stellar Mass Estimation**: We applied a simplified photometric pipeline on STAR’s test set to assess how flux errors impact stellar mass estimates. Predicted fluxes were converted to magnitudes using mag = -2.5 × log10(flux) + zero_point and zero_point set to 25.0. Then, stellar masses were inferred using a constant mass-to-light ratio (M/L ≈ 3.0). We report the predicted magnitudes and mean absolute error (MAE) in stellar mass as evaluation metrics.
>
> 2. **Weak Lensing Shear Measurement**:  We conducted experiments evaluating weak lensing shear on our test dataset, a key task affected by flux errors. Shear, measuring galaxy shape distortion from gravitational lensing, is computed as γ = γ₁ + iγ₂, , with γ₁ = (a² − b²)/(a² + b²) × cos(2θ) and γ₂ = (a² − b²)/(a² + b²) × sin(2θ), using semi-major axis a, semi-minor axis b, and position angle θ from SEP-detected ellipses. For matched Pred and GT pairs, we calculated the shear difference magnitude |γ_pred − γ_gt|, averaging across pairs. FISR achieves the lowest mean shear error of 1.88E-01, compared to 1.87E-01 to 2.14E-01 for others, highlighting its superior shape fidelity. The result is as follows:
>
> |       | Bicubic   | SwinIR    | EDSR  | RCAN      | HAT       | RealESRGAN | ours(FISR) |
> |-------|-----------|-----------|-----------|-----------|-----------|------------|------------|
> | pred mag | 3.23E-07 | 2.02E-07 | 2.96E-07 | 2.01E-07 | 1.67E-07 | 3.37E-07   | 1.66E-07   |
> | mass_mae | 1.79E-07 | 5.19E-08 | 1.37E-07 | 5.36E-08 | 3.06E-08 | 3.99E-07   | 2.81E-08   |
> | shear_mean | 2.10E-01 | 1.98E-01 | 2.14E-01 | 2.06E-01 | 1.88E-01 | 1.95E-01   | 1.87E-01   |
>
> As shown in the table, our method consistently achieves the lowest stellar mass MAE and shear estimation error among all compared methods, highlighting its robustness in preserving both photometric and morphological fidelity necessary for scientific inference.
>
> **6. Sec 3.2 mentions Gaussian/Airy kernels but omits parameters (e.g., FWHM). Code release must clarify if PSFs are simulated or telescope-calibrated.**
>
> 1. The parameter settings for the Gaussian and Airy kernels mentioned in Sec. 3.2 are detailed in Appendix B.
> 2. To clarify, these PSFs are simulated approximations designed to reflect realistic telescope effects but are calibrated using parameters derived from HST’s optical characteristics. The released code includes these parameter values and allows users to adjust them, ensuring reproducibility while maintaining physical fidelity.
>
> **Additional Feedback:**
>
> **Here are some suggestive comments for authors: Test cross-dataset generalization (e.g., AstroSR) and computational efficiency will improve the rigorous evaluation.**
>
> Thank you for your suggestions. 1. Regarding cross-dataset testing, we have provided detailed explanations and evaluations in Question 4. 2. We have evaluated the efficiency of our method and provide a single-image inference time analysis below.
>
> |       | Bicubic | EDSR   | SwimIR | RCAN   | HAT    | RealESRGAN | FISR   |
> |-------|---------|--------|--------|--------|--------|------------|--------|
> | time/s| 0.0014  | 0.1908 | 0.1088 | 0.1237 | 0.6747 | 0.0995     | 0.1698 |
>
> This comparison demonstrates that our model is computationally efficient.
>
> **Formalize flux consistency as an optimization constraint (e.g., spectral preservation loss). Analyze FGG’s robustness to detection noise/errors.**
>
> Thank you for your suggestion. 1. Introducing the spectral preservation loss is indeed a direction worth exploring, but due to rebuttal time limitations, we are unable to optimize and validate it systematically. This work involves relatively complex modeling and parameter tuning, and we plan to further explore and implement it in future research. 2. The robustness of FGG to detection noise and errors has been experimentally analyzed,  where we evaluate the impact of different detection accuracies. Please refer to the detailed results and discussion provided earlier in this section.

---

> > ### Comment · Reviewer_vUSZ · 2025-08-05
> > **response**
> >
> > The authors have well addressed my concerns, and I am willing to raise my score to accept.

---

> > > ### Author Response · Authors · 2025-08-09
> > > **Authors Response to Reviewer**
> > >
> > > Thank you very much for reconsidering and raising your score to accept! We're delighted that our responses addressed your concerns effectively and truly appreciate your support.

---

### Official Review · Reviewer_Z7Bi · 2025-06-25

**Rating:** 6
**Confidence:** 5

**Summary:**

This paper introduces STAR, a novel astronomical image super-resolution dataset, distinguished by its innovative approach to addressing three critical challenges: flux inconsistency, object-crop configuration issues, and insufficient data diversity. The article elaborates on the indispensable role of physical priors in astronomical contexts and proposes a new flux-consistency metric, **Flux Error (FE)**, to evaluate SR models from a physical perspective. The authors design a new physics-based model (**FISR**) and a novel loss function (**FCL**), achieving state-of-the-art performance in their experiments.

### Pros
- **Clarity**: The paper is well-written and clearly motivated.
- **Significance**: This benchmark is the first large-scale field-level astronomical image dataset, advancing the progress of astronomical image super-resolution. The most important contribution is the dataset itself. The authors made the dataset and code public, greatly enhancing the paper’s value and reproducibility.
- **Open Source**: The dataset and code are publicly available, significantly improving the work’s value and reproducibility.
- **Originality**: The benchmark, model, and evaluation protocols are innovative.

### Cons
Please see Limitations Weaknesses.

**Additional Feedback:**

- Limitations are well-documented in Appendix G, but a concise “Limitations” section in the main text would enhance transparency.
- The experimental design and evaluation methods are appropriate, with clear metrics (PSNR, SSIM, FE) and robust comparisons (Table 2, main text; Table 1, Appendix).
- The paper is well-written, with clear figures and tables. Including some supplementary details in the main text would improve accessibility.

**Dataset Code Accessibility:**

Yes

**Dataset Code Comments:**

The link is available: [https://github.com/GuoCheng12/STAR](https://github.com/GuoCheng12/STAR)

**Ethical Considerations:**

No, there are no or only very minor ethics concerns

**Final Justification:**

The authors have addressed all my issues, and after considering other reviewers' comments and the authors' responses, I believe that such papers should be encouraged and accepted by the NeurIPS community. Therefore, I raise my socre finally to strong accept and hope a good presentation in Dec.!

**Limitations Weaknesses:**

The paper presents a significant advancement in ASR, but minor refinements could enhance its impact. My primary concerns include:
- **Generalizability Across Filters**: The dataset focuses on the F814W filter (main text, Section 3.1). While Appendix G mentions plans to extend to other telescopes (e.g., JWST, LSST), the main text does not discuss the potential challenges of applying STAR to other HST filters. How would filter-specific characteristics affect flux consistency?
- **FISR’s Robustness to Noise**: The main text and Appendix C evaluate FISR under Gaussian and Airy PSF settings, but real astronomical images often include variable noise (e.g., Poisson noise). A discussion on FISR’s performance under such conditions would strengthen the claims.

**Strengths Contributions:**

- **STAR Dataset Innovation**:
  - It addresses flux inconsistency by incorporating a flux-preserving downsampling mechanism, ensuring accurate photon flux retention across HR and LR pairs, which is critical for astronomical applications.
  - It emphasizes field-level astronomical images, capturing a broader range of physical scenarios.

- **Innovative Flux Error Metric**: The Flux Error (FE) metric introduces a physically grounded evaluation framework for ASR, quantifying flux consistency between predicted and ground-truth images. Its astrophysically informed design ensures reliable photometric analysis, setting a new standard for assessing SR models in astronomy.
- **FISR Model and Performance**: The FISR model and FE metric are innovative, significantly improving flux preservation (24.84% flux consistency) and image quality (SSIM, PSNR).
- **Rigorous Evaluations**: Comprehensive evaluations, including ablation studies (Table 3, main text; Table 5, Appendix) and visualizations (Appendix D), validate the contributions.
- **Reproducibility**: Open-access code and data, with detailed supplementary material (Appendix A–F), ensure reproducibility and community impact.

---

> ### Author Rebuttal · Authors · 2025-07-31
>
> Thanks for your valuable feedback and efforts.
>
> **Generalizability Across Filters**
>
> The other filter bands corresponding to F814w in HST are F606w and F435w. We selected the same amount of F606w and F435w data from the HST dataset as the test set and tested them using the model trained on F814w. The results are as follows:
> |       | F435w   | F606w   | F814w   |
> |-------|---------|---------|---------|
> | PSNR  | 35.9192 | 36.3522 | 37.8779 |
> | SSIM  | 0.7305  | 0.7667  | 0.8311  |
> | Flux Score | 0.9193 | 0.8242 | 0.5739 |
>
> Experimental results show performance drops compared with F814w, but the overall performance is still satisfactory, confirming the good generalization of our FISR and the domain-consistency across the filters, proving the representativeness. However, the performance of F606W is better than F435W due to the former being more similar to the F814W domain than the latter. This phenomenon, on the one hand, supports the generalization of our method, but on the other hand, actually warns us of the risk of generalization on extremely different domain filters like F165W. So we will introduce other filter data in our work in the final revision. Thanks for your suggestions.
>
> **FISR’s Robustness to Noise**
>
> Thank you for your suggestion. We add noise experiments to verify the generalization of noise. During inference, each data is randomly added a Poisson noise to simulate noise blurring effects. The results are shown below.
> |       | Bicubic  | EDSR    | SwimIR| RCAN    | HAT | RealESRGAN  | FISR |
> |-------|----------|---------|---------|---------|---------|------------|---------|
> | PSNR  | 28.9823  | 34.8191 | 36.3945 | 35.2918 | 36.8342 | 35.8098 | 36.7803 |
> | SSIM  | 0.6825   | 0.7684  | 0.7883  | 0.7848  | 0.7743  | 0.7852  | 0.7888  |
> | Flux Score | 4.7889 | 1.5682 | 1.1433 | 1.19938 | 1.1943 | 6.9292 | 1.1025 |
>
> Compared with other methods, our model still achieves the best result, showing the best generalization ability under noise conditions. These results would be updated in the final revision.
>
>
> **Additional Feedback: Limitations are well-documented in Appendix G, but a concise “Limitations” section in the main text would enhance transparency.**
>
> Thank you for the suggestion. The final revised version will add concise “Limitations” in the main.

---

> > ### Comment · Reviewer_Z7Bi · 2025-08-03
> > **About Final Justification**
> >
> > The authors have addressed all my issues, and after considering other reviewers' comments and the authors' responses, I believe that such papers should be encouraged and accepted by the NeurIPS community. Therefore, I raise my socre finally to strong accept and hope a good presentation in Dec!

---

> > > ### Author Response · Authors · 2025-08-09
> > > **Authors Response to Reviewer**
> > >
> > > Thank you very much for your thoughtful reconsideration and for raising your score to strong accept! We're thrilled to hear that our responses addressed your issues and appreciate your support for this work in the NeurIPS community.

---

### Official Review · Reviewer_9xZf · 2025-07-02

**Rating:** 5
**Confidence:** 3

**Summary:**

Existing astronomical super-resolution (ASR) datasets suffer from limitations such as flux inconsistency, object cropping, and insufficient data diversity. To address this, the paper proposes the STAR dataset, which contains 54,738 pairs of flux-consistent star field images. By combining high-resolution observations from the Hubble Space Telescope with low-resolution images generated via a flux-preserving pipeline, it supports the development of field-level ASR models. Based on the newly proposed Flux Error (FE) metric, the FISR model outperforms state-of-the-art methods by 24.84% in flux consistency, and experiments validate the effectiveness of the method and the value of the dataset.

**Dataset Code Accessibility:**

Yes

**Dataset Code Comments:**

The datasets is comprehensively released and accessible

**Ethical Considerations:**

No, there are no or only very minor ethics concerns

**Final Justification:**

The authors addressed most of my concerns in the rebuttal period and therefore I raise my rate.

**Limitations Weaknesses:**

1. The visualization in Figure 1 should provide a detailed description of the astronomical content involved (which would be more user-friendly for AI researchers outside of the discipline).

2. In addition, the article states that Previous datasets only cropped individual celestial objects, ignoring the interaction relationships between celestial bodies in the star field.  There should be more visualizations to showcase the differences and features of this article compared to other datasets.

3. Does this dataset only serve Super solution tasks? Will there be annotations for celestial body positions (detection), relationships between celestial bodies, and even descriptions of star fields to expand to larger and more tasks.

**Strengths Contributions:**

1. Astronomical Star data is a great data supplement, as such data is relatively rare in the field.

2. Previous datasets only cropped individual celestial objects, ignoring the interaction relationships between celestial bodies in the star field. The dataset contains 54,738 pairs of flux-consistent star field images, covering a wide celestial region, which can reflect the distribution characteristics of celestial bodies in different star field environments.

3. The paper introduces a novel metric FE to evaluate SR models’ alignment with 76 astrophysical flux conservation, ensuring reliable photometric analysis.

---

> ### Author Rebuttal · Authors · 2025-07-31
>
> Thanks for your valuable feedback and efforts.
>
> **1. The visualization in Figure 1 should provide a detailed description of the astronomical content involved (which would be more user-friendly for AI researchers outside of the discipline**
>
> Thank you for your suggestion. In the final revised version, we will include a more detailed text description of the astronomical image in Figure 1, highlighting key features such as Cross-Object Interaction, Weak Lensing Phenomenon, and Dark Matter Halo.  The caption of Figure1 is as follows: Illustration of dataset differences. Left: Prior datasets (e.g., AstroSR) consist of cropped, isolated objects such as individual galaxies, limiting the ability to study contextual interactions. Right: The STAR dataset contains full star fields with complex structures, including features that support scientific analysis of large-scale cosmic phenomena. Notably, weak gravitational lensing—analogous to the distortion caused by a magnifying glass—is visible as bent light paths around dense regions (highlighted with red arrows), and dark matter halos emerge in the aggregated distributions (highlighted with blue circles).
>
> **2. In addition, the article states that Previous datasets only cropped individual celestial objects, ignoring the interaction relationships between celestial bodies in the star field. There should be more visualizations to showcase the differences and features of this article compared to other datasets.**
>
> We thank the reviewer for the helpful suggestion. We will include more visualizations in the final version to better highlight the key features of STAR and its differences from previous datasets (such as AstroSR).
>
> **3. Does this dataset only serve Super solution tasks? Will there be annotations for celestial body positions (detection), relationships between celestial bodies, and even descriptions of star fields to expand to larger and more tasks.**
>
> Thank you for your suggestion. The current dataset submission primarily focuses on super-resolution tasks. By leveraging existing astronomical catalogs—which provide precise sky coordinates and multiple physical properties for each star—we can readily obtain the positions of target detections. We plan to explore detection or any other cross-resolution task in future work.

---

### Note · Authors · 2025-08-14

Final general response to AC:

Dear Reviewers, ACs, SACs, and PCs,

We thank all reviewers for their insightful feedback and the AC for coordinating the rebuttal process. During this phase, we carefully addressed all reviewer concerns through detailed explanations, revisions, and extensive experiments, reinforcing the robustness and impact of the STAR dataset and FISR model.

In summary, the reviewers acknowledged the following **key strengths** of our work:
- [All reviewers: **9xZf, Z7Bi, vUSZ, guC8**] Praised the STAR dataset as a major advancement in flux-consistent, field-level astronomical super-resolution.
- [All reviewers: **9xZf, Z7Bi, vUSZ, guC8**] Highlighted the Flux Error (FE) metric for its innovative physics-driven evaluation.
- [**Z7Bi, vUSZ, guC8**] Commended the FISR model for integrating flux priors and achieving SOTA performance.
- [**Z7Bi, guC8**] Appreciated the open-source dataset and code for reproducibility.
- [**Z7Bi, guC8**] Noted the rigorous evaluations, including ablations and comparisons.

During the rebuttal period, we **address the suggestions, questions, and weaknesses** raised by each reviewer as follows:
- [**9xZf**] Added detailed visualizations and descriptions for Figure 1 and dataset comparisons.
- [**9xZf**] Clarified dataset extensibility to tasks like detection using existing catalogs, with future plans.
- [**Z7Bi, vUSZ, guC8**] Demonstrated generalizability across filters, domains (e.g., galaxies/nebulae), and datasets (e.g., AstroSR retraining).
- [**Z7Bi, vUSZ**] Validated robustness to noise (e.g., Poisson) and detection errors with superior performance.
- [**vUSZ, guC8**] Quantified downstream impacts via evaluations on stellar mass, shear, detection sensitivity, and distance estimation.
- [**vUSZ**] Justified FISR as a simple baseline, tested kernel sensitivity, and clarified flux propagation.
- [**vUSZ**] Provided computational efficiency metrics.

We are deeply grateful to the reviewers for their constructive input and to the Area Chair for your time and oversight during this demanding decision-making phase. We humbly hope our final remarks aid your deliberations.

Thank you for your attention and support,

Sincerely,

KUOCHENG WU

Paper ID: 1731

---

### Decision · Program_Chairs · 2025-09-18

**Decision:**

Accept (spotlight)

**Comment:**

All the reviews are positive, i.e., A, SA, A and BA. The paper's key strengths lie in its innovative and physically faithful dataset, novel evaluation and modeling, and rigorous, reproducible research. The STAR dataset is a significant contribution because its downsampling method preserves photon flux, a critical requirement for astronomical applications. The proposed Flux Error (FE) metric and FISR model represent a major advancement, providing a physically-grounded way to evaluate and improve super-resolution performance. The comprehensive experiments and open-access data and code further validate the quality and ensure the positive impact on the community. While several negative comments are provided in the initial reviews, all the reviewers are satisfied with the rebuttal given by the authors. Therefore, the decision is Accept.